# Structural Violence and the Effects of the Patriarchal Structure on the Diagnosis of Borderline Personality Disorder (BDP): A Critical Study Using Tools on BPD Symptoms and Social Violence

**DOI:** 10.3390/ijerph22020196

**Published:** 2025-01-29

**Authors:** Elena Valero, Alicia Paillet, Victor Ciudad-Fernández, Marta E. Aparicio-García

**Affiliations:** 1Instituto de Investigaciones Feministas, INSTIFEM, Universidad Complutense de Madrid, 28008 Madrid, Spain; evaler02@ucm.es (E.V.); alpaille@ucm.es (A.P.); 2Department of Personality, Evaluation and Psychological Treatments, University of Valencia, 46010 Valencia, Spain; viciufer@alumni.uv.es

**Keywords:** borderline personality disorder, social violence, gender stereotypes, emotional intensity, feminist critique

## Abstract

This study explores the relationship between borderline personality disorder (BPD) symptoms, measured using the Borderline Symptom List (BSL-23), and experiences of covert social violence, assessed via the Inventory of Covert Social Violence Against Women (IVISEM) and an open-ended survey given to 99 adults diagnosed with BPD. Quantitative data revealed significant emotional intensity, with a mean BSL-23 score of 56.81 (SD = 20.31), and a positive correlation (r = 0.29, *p* < 0.0034) between symptom severity and the number of self-reported disorders. The qualitative analysis highlighted themes of ‘Stigmatization and Structural Violence’ and ‘Gender Expectations’, with 62.9% of participants reporting that their emotions were pathologized as hormonal or exaggerated. The results highlight the significant emotional intensity in participants, particularly related to shame and vulnerability, suggesting these emotions are linked to structural violence perpetuated by patriarchal norms, including covert social violence. Biological explanations for emotionality, such as references to “hormonal” changes and “menstruation”, reinforce the idea that women’s intense emotions are natural, overlooking broader societal and structural factors. The results underscore the impact of the patriarchal structure, emphasizing the need for psychological approaches that address both the symptoms of BPD and the impact of societal and structural violence on women’s emotional health. The study sample underscores the main idea of the study: BPD is predominantly diagnosed in women, which underlines the need to rethink diagnostic tools and professional interventions. These results highlight the need for a feminist critique of the BSL-23 by showing how emotional symptoms are often interpreted through a gendered lens, emphasizing the importance of re-evaluating diagnostic tools to address the impact of societal and structural violence on women’s mental health.

## 1. Introduction

Borderline personality disorder (BPD) is predominantly diagnosed in women, accounting for approximately 75% of cases [1] Most of the research concludes that there is a gender bias in BPD diagnoses, as clinical professionals tend to diagnose this disorder more frequently in women [2]. This significant gender difference has been criticized from a feminist perspective, which argues that the diagnosis of BPD may be influenced by gender stereotypes that pathologize behaviors socially attributed to women, such as intense emotionality and instability in relationships [3,4] both in society and among diagnosing professionals. These characteristics, commonly associated with BPD, reinforce perceptions that women are “overly emotional” or “unstable”, whereas similar traits in men are often interpreted differently or even normalized [3,5]. Marcie Kaplan (1983) noted that the third edition of the Diagnostic and Statistical Manual of Mental Disorders (DSM-III) was predominantly written by men [6]. She argued that these authors incorporated gender biases into the creation of diagnostic categories [7] Kaplan explained that the group of professionals who worked on the DSM-III (37 men and 3 women) used characteristics aligned with masculine patterns to categorize illness and health [2]. As a result, women faced a higher risk of being diagnosed with histrionic or dependent personality traits due to the application of these gender stereotypes [6].

Another key aspect of the BPD diagnosis that reinforces a gender bias is the criterion related to inappropriate anger. This criterion has been critiqued in the literature for its formulation, which offers limited guidance in distinguishing between “appropriate” and “inappropriate” anger [3] (Dodd, 2015). Such ambiguity allows for interpretations influenced by deeply rooted gender stereotypes, both in individuals experiencing anger and in professionals evaluating these symptoms [5]. According to the studies, this could perpetuate a pathologization of behaviors that deviate from socially accepted norms of femininity, particularly for women diagnosed with BPD. Psychological research suggests that this criterion can be understood and applied in various ways, none of which are mandatory or exclusive. This ambiguity, combined with the working conditions in clinical settings, increases the risk of professionals relying on stereotypes to assess anger [5]. Specifically, empirical studies on public administration and management point to the widespread influence of gender stereotypes, where anger is traditionally associated with men, while women expressing anger are often perceived as emotionally unstable or out of control [5]. This could contribute to biases in diagnoses like BPD, where interpretations of anger may vary depending on the individual’s gender within therapeutic contexts.

Moreover, women who express anger are often seen as deviating from prescribed gender roles, and their anger is thus more frequently labeled as “inappropriate” [5]. In psychiatry, this tendency could contribute to women being more frequently diagnosed with BPD, as their anger might be more readily interpreted as pathological. This systematic bias reinforces the pathologization of female emotion and the unequal interpretation of the same symptoms in men and women [5]. In the 1980s, feminist protests against the DSM highlighted how certain psychiatric diagnoses included in the manual were sexist and pathologized feminine behaviors, reflecting broader social issues rather than individual disorders. Diagnoses like Masochistic Personality Disorder or Premenstrual Dysphoric Disorder reflected a patriarchal perspective that ignored the effects of gender socialization on women’s mental health. This critique remains central to the current debate on BPD, a diagnosis disproportionately applied to women and reinforcing gender stereotypes about excessive emotionality [3]. Feminists argued that many DSM diagnoses contribute to the pathologization of feminine behavior, presenting emotional problems as inherent to women’s nature. This dynamic is particularly evident in the interpretation of BPD symptoms, where traits like emotional instability or impulsivity are often framed within a diagnostic context for women, while violent or aggressive behaviors in men may be addressed through social or legal frameworks rather than as mental health concerns. These patterns reflect broader gendered dynamics that influence how emotional expression is evaluated and pathologized [8,9]. This underscores the importance of critically examining diagnostic tools like the BSL-23, which align closely with DSM-5 criteria, to ensure they do not perpetuate such biases in the assessment and treatment of BPD. This approach underscores the need to review tools like the BSL-23, whose items on self-harm or impulsivity may perpetuate a gender-biased perspective that disproportionately pathologizes women [3]. Feminist critiques also emphasize how the DSM differently interprets male and female behaviors: while male violence may be seen as a “normal” response to stress, sadness or anger in women is frequently labeled as pathological. This highlights the importance of considering sociocultural influences on female behavior and questioning how scales like the BSL-23 may reinforce biased narratives around BPD [3]. Finally, some feminists proposed alternative diagnoses that reflect the impact of male violence on women’s mental health, such as Battered Woman Syndrome [3]. This proposal sheds light on how social problems and power dynamics influence mental health—an approach that can also be applied to BPD. Thus, BPD should not be considered solely an individual problem but a reflection of socialization and patriarchal structures that affect women in our society [3].

In the psychiatric literature, a broad analysis of the symptoms of BPD has been carried out, highlighting the importance of grouping the symptoms into key dimensions that facilitate the clinical understanding and treatment of this disorder, as we, as professionals, are currently faced with a wide spectrum of symptoms. In the work of Clarkin et al. (2007), the symptoms of BPD are grouped into three major dimensions, affective dysregulation, interpersonal difficulties, and impulsivity, which constitute the core of the disorder [10]. However, from a feminist perspective, it is essential to question how these symptoms may have been pathologized differently in women, fueling gender stereotypes and perpetuating a biased view of female mental health. Difficulties in emotional regulation are a central feature of individuals with BPD. These difficulties are associated with temperaments such as high neuroticism and low extraversion, both commonly observed in patients with this disorder. The inability to adequately manage negative emotions may lead to increased impulsivity and self-destructive behaviors. This finding is particularly relevant when considering how gender stereotypes and covert social victimization can exacerbate these emotional difficulties in women and others socialized under restrictive gender expectations. Socialization based on emotional control and submission may heighten emotional reactivity, reinforcing patterns of affective instability associated with BPD [11]. Additionally, it has been found that individuals with BPD exhibit a reduced attentional bias towards positive stimuli, meaning they tend to focus less on stimuli that could generate positive emotional responses. This reduction in positive attentional bias is closely related to difficulties in regulating emotions, a central aspect of BPD. The lack of attention to positive stimuli can worsen emotional instability, as individuals with BPD do not benefit from stimuli that, in others, would help stabilize their emotional state. This mechanism is further affected in contexts of covert social victimization and gender stereotypes, where individuals with BPD, especially women, are exposed to environments that reinforce emotional negativity and increase negative emotional reactivity [12].

In recent years, various data suggest that women experience specific health problems that may stem from structural inequalities between men and women [13,14,15]. Studies in the health field have identified a link between the internalization of certain gender stereotypes and poorer health outcomes for women [16,17,18,19]. Covert social violence has been identified as a form of victimization with a widespread impact, affecting both psychological health and women’s perceptions of their societal roles [20,21]. In the context of BPD, this type of violence contributes to the pathologization of emotionality in women, such as shame and vulnerability, thus perpetuating stereotypes that stigmatize women’s behaviors as excessively emotional or unstable. Patriarchal structures, understood as social, cultural, and economic systems that perpetuate gender inequalities and power imbalances, provide the framework for the interpretation of symptoms such as anger, emotional instability, and impulsivity in women. These perverse structures contribute to the stigmatization of behaviors associated with this disorder, reinforcing biased diagnostic practices and limiting the understanding of the broader social and structural factors that influence women’s mental health.

This article seeks to connect this perspective with a BPD diagnosis, exploring how these experiences of covert violence impact symptom severity and emphasizing the need to address the influence of patriarchal structures. Patriarchal structures, understood as social, cultural, and economic systems that perpetuate gender-based inequalities and power imbalances, shape the interpretation of symptoms like anger, emotional instability, and impulsivity in women with BPD. These structures contribute to the stigmatization of behaviors associated with the disorder, reinforcing biased diagnostic practices and limiting the understanding of the broader societal and structural factors influencing women’s mental health.

This study examines the intersection of covert social violence, patriarchal structures, and BPD symptoms to explore how societal norms, and systemic inequalities shape the interpretation and pathologization of feminine emotionality. Building on prior research, this approach emphasizes the need to critically evaluate diagnostic tools like the BSL-23, which may reflect and reinforce gendered biases in mental health diagnoses. Previous studies have analyzed the influence of gender stereotypes and structural violence on psychiatric diagnoses, often through theoretical or quantitative methods [8,9]. In contrast, this study incorporates a qualitative analysis to amplify the voices of the affected women, providing a nuanced perspective that connects structural critiques with personal experiences. This methodological choice aims to address identified gaps in the literature, offering a more comprehensive understanding of how gender dynamics influence mental health outcomes.

## 2. Materials and Methods

### 2.1. Participants

The participant sample consisted of a total of 99 individuals. The mean age was 29.64 years (SD = 7.96) with a median age of 28 years (MAD = 7.41). The majority of participants were cisgender women (88.89%), reflecting the gendered nature of the condition under study, were single (31.31%), were of heterosexual orientation (30.1%), had completed secondary education (48.48%), were employed (33.33%), reported their financial situation as comfortable (52.23%), and had two additional disorders (43.43%). Further details are available in Table 1.

### 2.2. Measures

Sociodemographic measures. Participants answered a set of questions regarding age, gender, perception of being considered a racialized person, nationality, civil status, educational level achieved, employment status, economic status, sexual orientation, and the number of additional psychological disorders (besides BPD).

Borderline Symptom List-23. This instrument was developed to assess the severity of BPD [22]. For this study, the brief Spanish validation was employed [23]. This psychometric tool comprises 23 items (e.g., “The criticism had a devastating effect on me”). Items are scored on a 5-point scale (0 = Not at all; 4 = Very strong). The latent structure is composed of a single factor. The Spanish validation reported excellent reliability (α = 0.94). In the present study, similar internal consistency was found (α = 0.94; ω = 0.96). The BSL-23 scale was chosen because it is a validated and reliable tool for assessing the severity of BPD symptoms, focusing on core aspects of the disorder such as emotional dysregulation and impulsivity. One of its key strengths is its simplicity: with just 23 items, it is brief, easy to understand, and practical to use in research and clinical settings. The items are closely aligned with the symptom criteria in the DSM-5, which makes it a direct and accessible measure for exploring the traits commonly associated with BPD. Additionally, the BSL-23 has been validated in Spanish and demonstrates excellent reliability, which was particularly important for this study. From a feminist perspective, this scale is relevant because it reflects the same diagnostic framework that has been criticized for pathologizing traits like emotional intensity and impulsivity—characteristics often attributed to women. By using the BSL-23, this study seeks to examine these symptoms through a feminist lens and challenge the gender biases that may be embedded in traditional diagnostic tools.

Inventory of Covert Social Violence Against Women (IVISEM). This scale was used to measure different dimensions of gender mandates that women assume as something that is normalized and that, in some way, subject them to the male figure, resulting in a form of socially accepted victimization. The questionnaire was developed originally in Spanish [24]. It comprises 35 items comprising seven subscales and one second-order factor. The subscales are as follows: Maternity (e.g., Mothers have a special bond with their children that fathers do not have), Romantic love and partner (e.g., The ideal is to find a partner to be happy with forever), Care (e.g., When children need to be taken to the doctor, mothers understand and follow the instructions better than fathers), Career projection (e.g., If a choice must be made between the woman and the man to care for the children, it is more convenient for the woman to give up part of her professional life), Attitudes and submission (e.g., Men are usually the ones who make important financial decisions), Biology and abilities (e.g., In general, women have worse spatial abilities. For example, they are worse at reading maps), Neosexism (e.g., In reality, feminists only seek equality, not the superiority of women over men). The validation reported a good fit for the heptafactorial model and excellent internal consistency for the general factor (α = 0.93; ω = 0.95). In the present study, internal consistency ranged from acceptable to excellent across subscales: Maternity (α = 0.59; ω = 0.70); Romantic love and partner (α = 0.85; ω = 0.90); Care (α = 0.82; ω = 0.88); Career projection (α = 0.75; ω = 0.87); Attitudes and submission (α = 0.83; ω = 0.89); Biology and abilities (α = 0.76; ω = 0.86); and Neosexism (α = 0.88; ω = 0.92). Finally, the general scale showed excellent internal consistency (α = 0.94; ω = 0.96). The IVISEM scale was included to explore covert social violence—a form of structural victimization rooted in gender norms and societal expectations. This scale assesses dimensions like caregiving, romantic relationships, and submission, which are particularly relevant to understanding how patriarchal structures impact women’s psychological experiences. It offers a way to analyze how these social pressures shape emotional and mental health, aligning with feminist theories that emphasize the role of structural violence. Including the IVISEM scale allows this study to connect the symptoms of BPD with broader societal factors, providing a deeper understanding of how gendered experiences influence mental health outcomes.

Experiences with Gender, Stigma, and Diagnosis Survey. Participants completed a custom-designed survey to assess their experiences related to the intersection of gender, stigma, and borderline personality disorder (BPD). The survey included items that explore the perceived misinterpretation of symptoms based on gender, experiences of stigma and discrimination, gender-based modifications in treatment, exposure to violence (e.g., physical and sexual abuse), social pressures related to traditional gender roles, and the impact of the BPD diagnosis on their emotional well-being and personal achievements.

Qualitative measures. To explore participants’ subjective experiences with borderline personality disorder (BPD) and the influence of gender stereotypes, an open-ended questionnaire was designed. The qualitative section aimed to gather in-depth insights into the following themes: Diagnostic Journey: Participants were asked about their experiences with the diagnosis of BPD, including who provided the diagnosis, how their symptoms were interpreted, and whether these interpretations were influenced by gender. Stigma and Discrimination: The questions explored experiences of stigma, such as being blamed for emotional difficulties due to gender stereotypes, and instances of discrimination related to their diagnosis or gender. Trauma and Gender Expectations: Participants were asked about their history of trauma and whether societal expectations of gender roles impacted their emotional well-being and interpersonal relationships. Emotional Intensity and Gender Bias: Participants reflected on how gender stereotypes influenced the interpretation and management of intense emotions or anger by mental health professionals. Violence and Structural Norms: The questions addressed the role of societal norms in shaping participants’ interpersonal and emotional experiences, as well as their interactions in medical and therapeutic contexts. An example question was as follows: “Do you think gender stereotypes have influenced how your intense emotions or episodes of anger are interpreted?”.

This study integrates quantitative and qualitative methods in order to explore the intersection of gender, structural violence, and borderline personality disorder (BPD). Quantitative tools such as the BSL-23 and IVISEM provide measurable information, with validated scales, on symptom severity and the influence of covert social violence, while qualitative open-ended questionnaire data capture the lived experiences and nuanced perspectives of participants. This mixed-methods approach aligns with Creswell’s (2008) assertion that combining quantitative and qualitative methodologies allows researchers to address complex questions by integrating measurable patterns with personal narratives [25]. Furthermore, this approach bridges the limitations of each method, creating a more comprehensive understanding of how structural and social factors shape the experiences of women with BPD. By reducing the distance between quantitative data and qualitative insights, the study offers a holistic framework that situates individual experiences within broader societal contexts, echoing the importance of triangulation strategies highlighted [26].

### 2.3. Data Analysis

#### 2.3.1. Data Analysis of Quantitative Research

First, to describe the sample, frequencies and percentages were calculated for categorical sociodemographic variables, and the mean, median, standard deviation, and median absolute deviation were computed for age. Second, to describe the sample’s questionnaire scores, several estimates were calculated. Specifically, the mean, standard deviation, median, median absolute deviation, minimum, maximum, skewness, and kurtosis were computed for all items on the BSL and IVISEM scales, as well as for the total score of the BSL scale and the scores of all IVISEM subscales.

Third, the internal consistency reliability was assessed using Cronbach’s α and McDonald’s ω for each dimension of the BSL and IVISEM scales. Given the ordinal nature of the items (i.e., Likert-type scales), polychoric correlation matrices were employed to compute these reliability coefficients, as they are more appropriate than Pearson correlation matrices for ordinal data [27].

Fourth, the association between the total score on the BSL and the number of self-reported disorders (excluding BPD) was explored using Spearman’s correlation coefficient, given the non-normal distribution of the variables [28].

Given the sample size and exploratory nature of this study, the analyses chose to focus on descriptive, reliability, and correlational methods to identify measurable patterns in the data. These methods were complemented by thematic analyses to delve deeper into the nuanced ways in which structural and social factors intersect with participants’ experiences. Although more complex statistical models (e.g., multivariate regressions) could not be performed given the sample size, the mixed-methods approach provided a solid foundation for understanding the research questions.

#### 2.3.2. Data Analysis of Qualitative Data

The qualitative data collected through open-ended questions were analyzed using thematic analysis guidelines. The process began with a thorough review of the responses to familiarize researchers with the content, followed by systematic coding using NVivo 15 software to identify key ideas and patterns relevant to the study’s objectives. Codes were then grouped into broader themes, such as ‘Stigmatization and Structural Violence’, ‘Emotional Intensity and Gender Expectations’, ‘Guilt and Gender Roles’, ‘Impact of Diagnosis on Identity’, ‘Intersections between Violence and Gender’, and ‘Perceptions of Therapeutic Relationships’, which were reviewed and refined to ensure consistency and coherence. The final themes were defined and supported by excerpts from participants’ responses, and the qualitative findings were integrated with the quantitative results to provide a comprehensive understanding of the role of gender stereotypes and covert social violence in shaping participants’ experiences.

The integration of quantitative and qualitative findings was guided by the study’s objectives. Quantitative correlations from the BSL-23 scores regarding symptom severity and comorbidities informed qualitative themes such as ‘Emotional Intensity and Gender Expectations’ and ‘Perceptions of Therapeutic Relationships’. This approach allowed for a more nuanced understanding of how structural and social factors, including patriarchal structures, shape participants’ lived experiences. In addition to a construct-level analysis, item-level descriptive statistics were calculated to provide exploratory insights into the specific symptomatology and dimensions of covert social violence. This approach was intended to highlight nuanced patterns that may inform subsequent research or clinical applications.

## 3. Results

### 3.1. Quantitative Research

The analysis of individual items was included as an exploratory approach to offer deeper insights into specific symptoms and dimensions relevant to the study objectives. These findings complement the construct-level analysis provided by the total and subscale scores of the BSL-23 and IVISEM

The descriptive statistics for the items from the BSL scale, along with the total score, are presented in Table 2. For Item 13 (“I suffered from shame”), the mean score is 2.69 (SD = 1.31). Item 17 (“I felt vulnerable”) shows a higher mean score of 3.14 (SD = 1.08). For Item 23 (“I felt worthless”), the mean score is 2.63 (SD = 1.50). The total score on the BSL scale has a mean of 56.81 (SD = 20.31), with scores ranging from 8 to 92.

The descriptive statistics for selected items from the IVISEM scale are presented below in Table 3. Item 10 (“A woman takes better care of children and the elderly because she has a greater capacity for self-denial and sacrifice”) has a mean of 2.29 (SD = 1.48). Item 13 (“Women have more anxiety problems due to hormonal changes”) shows a higher mean (M = 3.05, SD = 1.38). Item 19 (“Women are usually more submissive than men”) has a mean of 2.23 (SD = 1.43). Item 26 (“It is normal for a girl to command more respect than a boy and to show more prudence in sexual behavior”) presents a mean of 2.67 (SD = 1.44). Item 33 (“It is more important for a woman to show prudence than for a man”) shows the lowest mean, at 1.72 (SD = 1.16). Lastly, Item 34 (“Women are more sensitive than men”) has a mean of 2.74 (SD = 1.49).

The IVISEM scale comprises several subscales, each designed to assess a distinct dimension. The Maternity subscale has a mean score of 13.32 (SD = 3.97). For the Romantic Love and Partner subscale, the mean is 11.14 (SD = 4.40). The Care subscale shows a mean score of 11.69 (SD = 5.01). The Career Projection subscale has a mean of 10.33 (SD = 4.15). The Submission Attitudes subscale presents a mean of 11.17 (SD = 4.81). The Biology and Abilities subscale has a mean score of 13.36 (SD = 4.70). The Neosexism subscale shows a lower mean score of 8.53 (SD = 4.48), with a range of 5 to 22. Finally, the Total Score across all subscales has a mean of 79.55 (SD = 23.68), with scores ranging from 41 to 142. Figure 1 displays the frequencies of the scales in tables, providing a detailed distribution of scores for each subscale.

Among the subscales, the Biology and Abilities and Maternity subscales show the highest mean scores. In contrast, the Neosexism subscale has the lowest mean score.

As Figure 2 shows, the Spearman correlation coefficient between the total score on the BSL scale and the number of self-reported disorders is 0.29, indicating a significant and positive relationship (*p* < 0.0034).

Additional data collected through the Experiences with Gender, Stigma, and Diagnosis Survey provided further insights into participants’ experiences with BPD and its intersection with gender. A total of 72.7% of participants reported receiving a BPD diagnosis between 2019 and 2024, with 60.8% stating that their symptoms had been misinterpreted due to gendered answers such as the following: “It is sexist and completely stigmatizing and stereotyped, with many comments such as “I don’t know if it’s because you’re a woman or because you have BPD”, or “I think you’re just a woman; women are naturally intense”. Additionally, 24.3% experienced gender-based changes or adjustments in their treatment, and 57.1% reported stigma-related experiences: “There is a lot of stigma, they tend to perceive us as manipulative and bad people, I think there is a lot of misinformation” or “That from the first word that it is disorder, a prejudice is made about others and ourselves, there is a certain discrimination for those who have “disorders”” or “They fired me from a job when I was 19 when they found out and then it happened again when I was 31, they fired me when they found out. It had been a while and until I found out, they were very happy with me”. Gender-specific blame for emotionality was reported by 62.2% of participants: “They make us look exaggerated or dramatic” or “People often make comments about me being crazy”.

Regarding experiences of violence and social pressure, 64.6% reported physical abuse by a partner or family member (“I have had partners who have used my disorder as a wild card to shield themselves”), 64.6% experienced sexual abuse (“That they treat me as crazy. And tell me that it is not my fault for being a “liar” (about sexual abuse) that it was because of my diagnosis”), and 59.6% reported intimate partner violence or domestic violence (“Generally, ex-partners who found out about my diagnosis greatly invalidated any opinion I had and used it to hurt me” or “My husband tells me that sometimes he would like to know what our marriage would be like if I were normal, he enjoys making me feel crazy”). Furthermore, 84.8% reported experiencing pressure to conform to beauty standards, while 74.7% highlighted societal pressure to adhere to traditional gender expectations: “People believe that just because I am a cis woman, the intensity in my relationships is normal, that I am erratic and hormonal”. Or “I have adopted caregiver roles that did not correspond to me just because I am a woman, I have been taught to treat partners as children” or “Well, at the beginning when I was a teenager, when I had boyfriends, I let them do whatever they wanted with me even if I didn’t want to until one day I said no and they forced me” or “I have felt pressure to fulfill certain roles or expectations, such as being more accommodating or emotionally available, which has affected the way I express myself and connect with others”.

In therapeutic contexts, 44.9% of participants reported differences in treatment based on gender. Additionally, 61.6% indicated that their BPD diagnosis had been used to invalidate their experiences (“When you explain your diagnosis, now I don’t usually say it, they treat me as dangerous to others. When the only danger is to myself”), 51.5% felt it was used to minimize their achievements or decisions (“I lost friends because they called me crazy, people took my ex-boyfriend’s side because they said he was right with me that I sleep with anyone and that it’s not normal”), and 62.9% noted stereotypes in the interpretation of intense emotions or anger (“We are considered hysterical for the simple fact of being a woman, all mood changes or impulses are associated with hormones” or “ Of course, because of the “hormonal”, especially when one is on their period. When you get angry they tell you to be hysterical or bitter”). Lastly, 27.6% encountered professionals with gender-based biases.

### 3.2. Qualitative Research

The qualitative analysis identified key thematic categories derived from participants’ responses, which are presented in Table 4. These categories were constructed based on the most frequent keywords and their respective counts and weighted percentages. The weighted percentages represent the proportion of each keyword relative to the total number of references across all categories. This approach ensures that less frequent but thematically significant keywords are not overshadowed by more common terms. The results provide a detailed representation of the linguistic patterns and thematic structures captured in the data, highlighting the centrality of gendered experiences, emotional expression, and diagnostic processes in the participants’ narratives. The thematic categories and keywords were derived using an iterative coding process grounded in thematic analysis guidelines, supported by NVivo 15 software. Keywords were selected based on their frequency and relevance to the study’s objectives, ensuring that central themes such as ‘Stigmatization and Violence’ and ‘Gender Norms’ were adequately captured. The thematic categories presented in this study correspond to nodes and subnodes generated in NVivo 15. This hierarchical coding system enabled the identification and organization of recurring themes and subthemes, ensuring that the analysis accurately reflected the participants’ narratives and captured both the breadth and depth of their experiences. The table offers a systematic summary of these findings, organized by thematic category and supported by the frequency and proportion of key terms identified in the analysis. Frequencies represent the total number of references coded under each thematic category, while weighted percentages indicate the proportion of each node relative to the total coded dataset. This dual metric highlights both the prevalence and contextual relevance of specific themes within the qualitative data.

A visual representation of the most frequently used keywords identified in the qualitative analysis is provided in Figure A1, illustrating the prominence of terms related to gender, emotions, and diagnostic experiences. For instance, while the term ‘stereotype’ appeared less frequently, it was central to discussions under the category ‘Gender Norms’, providing critical insights into the participants’ experiences with societal expectations.

The qualitative analysis identified key thematic patterns reflecting participants’ experiences with BPD and its intersections with gender norms and societal expectations:

Symptoms and experiences associated with BPD: Figure A2 presents the coding distribution of primary BPD symptoms. Intense anger (19 references) and suicidal behavior (12 references) emerged as the most frequently coded symptoms, followed by emotional instability and impulsivity. These results highlight the centrality of emotional dysregulation and self-harm in participants’ narratives.

General thematic categories: The distribution of references across general thematic categories is shown in Figure A3. Experiences of stigma and prejudice (192 references), opinions on the BPD label (181 references), and structural violence (145 references) were the most prominent themes, underscoring the pervasive influence of societal and systemic factors in shaping participants’ experiences.

Gender and BPD relationships: Figure A4 and Figure A5 illustrate the conceptual relationships between gender norms and BPD experiences. Participants frequently reported instances of pathologization and invalidation tied to gendered stereotypes, with the construct of the “crazy woman” emerging as a recurrent theme. These visualizations provide insight into how gender expectations influence both the interpretation of symptoms and therapeutic approaches.

Intense emotions and anger: Figure A6 focuses on coding related to intense emotions and anger. Subthemes included the interpretation of emotions through stereotypes (98 references), a gender bias in managing emotionality (90 references), and the expectations of gender in anger management (40 references). These results suggest a gendered lens in clinical settings that disproportionately pathologizes emotional intensity in women.

## 4. Discussion

This study makes a significant contribution to the existing literature by examining the intersection of gender and the diagnosis of borderline personality disorder (BPD) through a feminist lens. As highlighted in previous research, the diagnostic criteria for personality disorders, including BPD, are deeply influenced by cultural constructions of gender, revealing inherent biases in psychiatric classification systems [29]. These biases disproportionately pathologize traits such as emotional intensity and impulsivity in women, reinforcing patriarchal norms that dictate how emotions should be expressed according to gender. While some voices have begun to address these issues, it is essential that we go beyond reflective and revisionist critiques. This requires integrating individuals with BPD diagnoses into the knowledge production process and fostering longer-term initiatives that not only question existing frameworks but also identify and address specific shortcomings. By doing so, the field can move towards creating impactful changes that challenge systemic failures and promote more equitable diagnostic and therapeutic practices. Consistent with Creswell’s (2008) emphasis on the value of mixed methods, this research integrates quantitative tools, such as the BSL-23 and IVISEM, with qualitative insights in order to provide a holistic understanding of how structural violence and gender norms shape mental health outcomes [25]. Quantitative findings from the BSL-23 highlight the prevalence of emotional intensity, a trait disproportionately pathologized in women, while qualitative data reveal the lived experiences of stigma and misinterpretation rooted in gender stereotypes. While the quantitative findings from the BSL-23 and IVISEM scales provide information on the prevalence of emotional dysregulation and the impact of structural violence, the qualitative data complement these results by offering a deeper understanding of how these variables intersect with experiences lived by the participants. The qualitative responses help contextualize the statistical patterns observed in BSL-23 and IVISEM, particularly by revealing how gender stereotypes and structural violence are experienced at the individual level. In the results, we can see that, while the BSL-23 highlights emotional intensity as a common trait among participants, the qualitative data provide narratives that illustrate how these intense emotions are often pathologized due to gender stereotypes, such as being labeled as “hysterical” or “exaggerated”. Integrating both types of data strengthens interpretation by providing a holistic view of the complex factors that shape the experiences of people diagnosed with BPD. The integration of qualitative and quantitative data allows for triangulation, enhancing the validity of findings [26]. These results resonate with feminist critiques, which underscore the patriarchal underpinnings of diagnostic criteria that stigmatize traits associated with femininity [3]. The selection of the BSL-23 and IVISEM scales was grounded in their ability to capture the intersection of structural violence and patriarchal norms in mental health outcomes. The BSL-23 assesses traits like emotional dysregulation and impulsivity, which are frequently pathologized in women due to gendered expectations. The IVISEM, on the other hand, examines covert social violence through dimensions such as caregiving roles, submission, and romantic relationships, providing a broader understanding of how gender stereotypes influence psychological experiences. Together, these tools align with the study’s aim to critically explore the systemic biases embedded in diagnostic frameworks and their impact on women’s mental health.

Building on the lack of critical reviews of diagnostic manuals like the DSM, as highlighted earlier [6], these tools fail to adequately address cultural and gendered perspectives. This omission perpetuates structural inequalities by relying on diagnostic criteria that are shaped by patriarchal norms and exclude the lived experiences of women and other marginalized gender identities. These shortcomings not only reinforce the stigmatization of traits traditionally associated with femininity, such as emotional intensity, but also limit the capacity of mental health care systems to provide equitable and effective interventions. The absence of an intersectional and feminist critique in these manuals underscores the urgent need for a systematic revision of diagnostic frameworks to reflect diverse sociocultural contexts and experiences.

The descriptive results from the BSL-23 show that items related to intense emotions, such as shame and vulnerability, have high scores, indicating a high emotional intensity in the participants. The total mean suggests a moderate severity of BPD symptoms, but with significant variability in the expression of these symptoms, highlighting the importance of personalized therapeutic approaches that address both intense emotions and self-esteem, beyond the labels. This finding aligns with the critique presented by Dodd (2015), who emphasizes how intense emotions, commonly associated with women, are pathologized within the BPD diagnosis [3]. Throughout history, traits deemed desirable or undesirable have been assigned based on gender. Masculine traits have been framed as superior to feminine ones, with men associated with aggression and activity, and women with vulnerability, passivity, and caregiving roles [30]. These cultural constructions have been naturalized over time, rendering gender invisible and attributing human phenomena predominantly to biological sex [2]. Women who deviate from these roles often face pathologization, being labeled as “bad women”, “crazy”, or “mentally ill” [31,32]. This framework underpins the historical parallels between hysteria and BPD, where both serve as tools to label and control behaviors that challenge patriarchal expectations [33]. These dynamics underscore the urgent need to critically reassess the theoretical frameworks and diagnostic tools used in mental health care.

Although the results do not allow for a firm conclusion due to the sample size, there is a preliminary correlation between higher scores on the BSL-23 and the presence of comorbidities such as anxiety and depression. These findings reinforce previous studies suggesting that social and structural factors play a significant role in exacerbating mental disorders. As reflected in the BSL-23 scores, higher severity levels are associated with greater psychological distress and functional impairment. This correlation aligns with the observations of Vives-Cases et al. (2007), who highlighted how internalizing gender stereotypes can contribute to worse health outcomes for women [15]. Furthermore, this study supports the findings of Pérez (2021), who noted that psychiatric morbidity in women is more prevalent in depressive, anxious, or phobic disorders, as well as borderline, histrionic, and dependent personality disorders; whereas, in men, schizotypal, antisocial, and obsessive-compulsive personality disorders are more frequently observed [8]. The perspective of Shaw and Proctor (2005) highlights how gender dynamics have historically confined women to roles associated with emotionality and irrationality, particularly when they challenge traditional notions of femininity [9]. The DSM-5 reflects how gender stereotypes can become embedded in diagnostic criteria, leading to a biased interpretation of symptoms. This is evident in the frequent diagnosis of women with BPD and men with Post-Traumatic Stress Disorder (PTSD) [34]. This context underscores the relevance of choosing the BSL-23 for this study, given its close alignment with DSM-5 symptoms. By using this scale, the research aims to critically examine the gendered biases inherent in the diagnostic framework and explore how these biases influence the experiences and treatment of individuals diagnosed with BPD.

In the IVISEM scale, items related to motherhood and biological abilities show high scores, suggesting that participants internalize traditional gender expectations. In contrast, items related to neosexism show lower scores, indicating a general rejection of more modern ideas about gender inequality. These findings underline how gender stereotypes impact the emotional and psychological experience of participants, especially regarding the BPD diagnosis, supporting the idea that gender stereotypes contribute to the pathological interpretation of emotions in women diagnosed with BPD [3].

Qualitative data show that participants experience a pathologization of their identity through gender stereotypes, especially the label of “crazy” or “hysterical”. The intense emotionality of women is frequently misinterpreted and seen as a symptom of BPD due to gender stereotypes. This finding reinforces the idea that women are more likely to be diagnosed with BPD when their emotions are seen as overwhelming or uncontrollable, as pointed out in previous works [3].

A high percentage of participants reported experiencing physical and sexual violence, as well as social pressure to conform to beauty standards and traditional gender expectations. These social and structural factors are critical, as they exacerbate BPD symptoms and must be central to therapeutic interventions. Covert social violence has been identified as a significant factor of victimization with profound mental health implications, particularly for women [20]. Moreover, the higher prevalence of sexual abuse in girls compared to boys is a crucial factor to consider. This aligns with existing research that highlights the connection between structural violence and mental health outcomes, reinforcing the importance of addressing these factors in both research and clinical practice.

In therapeutic contexts, participants reported differences in treatment based on gender, with frequent invalidations of their experiences due to their BPD diagnosis. This underscores the need for gender-sensitive approaches to ensure women’s experiences are fairly considered without being reduced to socially constructed labels like “crazy”, “intense”, or “hysterical”. This finding aligns with feminist critiques, such as those by Dodd (2015), which highlight how BPD diagnoses often pathologize women’s emotional experiences and reinforce gendered expectations [3]. Similarly, Shaw and Proctor (2005) argued that BPD is a social and cultural construct rooted in norms of what is deemed “normal”, rather than objective criteria [9]. Psychiatry, they contended, has historically perpetuated gendered dynamics, often associating women with emotionality and irrationality, particularly when they challenge traditional femininity [35]. These critiques emphasize the urgent need to address the gender bias in both diagnostic and therapeutic practices.

Our findings resonate with feminist critiques of psychiatric diagnoses, emphasizing how gender stereotypes and structural violence disproportionately pathologize women. This study highlights the need to address systemic inequalities that influence the interpretation and diagnosis of mental health symptoms, reinforcing patriarchal biases in clinical settings. The diagnostic criteria for disorders like BPD often reflect societal expectations of women’s emotionality, framing traits like emotional intensity and impulsivity as pathological [3,5]. Similarly, empirical research underscores how structural gender inequalities shape mental health outcomes, reinforcing biases in diagnostic practices [13,15]. These critiques, which are deeply rooted in feminist and clinical perspectives, support the idea that tools like the DSM perpetuate patriarchal norms through diagnostic categories that disproportionately affect women. By integrating these insights, our study highlights the urgent need to reconsider diagnostic frameworks to address the sociocultural and structural context of women’s mental health. Although the findings suggest a relationship between structural factors and the exacerbation of BPD symptoms, it is important to emphasize that this study has a correlational and exploratory design. Therefore, these findings should be interpreted with caution, as they do not demonstrate a definitive causal relationship but rather an association between these variables.

This study has limitations that should be considered when interpreting the results. First, the sample size of 99 participants limits the statistical power of the analysis, which could affect the ability to generalize the results to a broader population that would not occur with a larger sample. While the item-level analysis offers valuable exploratory insights, it is acknowledged that these scales are primarily designed for a construct-level evaluation. Future studies could further validate these findings by focusing on broader patterns across constructs. The use of incidental sampling in this study represents a limitation to consider, as it may have introduced selection biases that affect the generalizability of the findings. Participants were primarily recruited through mental health professionals, activists, and social media, which may have resulted in an overrepresentation of individuals already involved in conversations about BPD or with access to these networks. Consequently, the sample might not fully reflect the diversity of experiences among individuals with BPD, particularly those who are outside these services, those experiencing high stigma who have not yet sought peer groups or verbalized their diagnosis, those who remain undiagnosed, or those with limited access to mental health resources. This limitation highlights the need for future research to use more systematic and representative sampling methods, such as stratified or random sampling, to ensure the greater applicability of the findings.

Another aspect to consider is that we are working with self-reported human experiences, which may be subject to memory or social desirability biases, particularly in the context of experiences of violence and stigmatization. Lastly, although a combination of quantitative and qualitative approaches was included, qualitative data are subjective and may not reflect the totality of the experiences of participants and people with BPD. The questions may not have been appropriate for the main objective of the study.

An important limitation in this study is the lack of male representation, particularly cisgender men, which prevents gender comparisons and highlights how BPD is still considered a predominantly female disorder. The absence of cisgender men in the sample suggests that BPD is a disorder “for and by” women. The lack of gender diversity in the sample highlights the need to rethink diagnostic tools and psychological interventions: “Where are the men with BPD? Are they being diagnosed appropriately? Do all the women who have responded have BPD? What is happening with the professionals who are diagnosing? Why have more than 70% of the diagnoses been in the last 6 years?”. Moreover, the frequent misdiagnosis of men with antisocial personality disorder instead of BPD may result in many not identifying with the disorder and, consequently, not participating in research or accessing questionnaires designed for BPD populations [2]. This misalignment underscores the necessity of re-evaluating diagnostic frameworks and ensuring that diagnostic tools are inclusive of diverse presentations of BPD across genders. Future research should incorporate specific recruitment strategies to include a more diverse representation of gender identities. Such diversity would allow for a broader understanding of the intersection between gender, stereotypes, and the diagnosis of BPD, enriching clinical practice and challenging gender biases in diagnostic processes.

It is important to note that potential biases in thematic grouping may arise during the coding process, despite the use of NVivo15 software. Although NVivo15 provides a systematic approach to coding, the subjective nature of qualitative analysis and researchers’ interpretation during response categorization may still influence the results. These biases could affect how themes are identified and affect the overall interpretation of the qualitative data. Future studies could mitigate these biases by using additional methods, such as peer review or intercoder reliability checks, to further validate the thematic analysis.

Future studies could address the limitations mentioned above by increasing the sample size and using random sampling methods to improve the representativeness of the sample and the external validity of the results. Including a more diverse sample, particularly in terms of gender, would provide valuable insights into how gender affects the experiences of people with BPD and other demographic characteristics across different social and cultural contexts. This study offers important perspectives on the experiences of cisgender women in the diagnostic processes of BPD; however, the predominance of cisgender women in both the sample and BPD diagnoses highlights a limitation. The study does not include the perspectives of men or individuals with diverse gender identities, restricting broader conclusions about how gender-based biases influence BPD diagnoses beyond cisgender women. Future research should prioritize the inclusion of men and individuals with non-cisgender identities to gain a more comprehensive understanding of how gender operates within psychiatric diagnostic frameworks. Expanding the sample diversity would also enable deeper analyses of the structural and systemic factors shaping mental health outcomes across genders. Previous research indicates that BPD diagnoses have a high prevalence in the LGTBQI+ population [36]. However, studies on trans and non-binary individuals remain scarce. Historical classifications, such as the DSM-III, included “doubt about gender identity” as a manifestation of “identity alteration” in BPD patients, which associated transgender experiences with a possible variant of the disorder [37]. Moreover, some studies suggest that behaviors associated with BPD may manifest differently in men, often leading to misdiagnoses such as antisocial personality disorder rather than BPD. These further underscores the need to explore how gender and diagnostic criteria intersect, as some experiences may not align with current diagnostic frameworks [2].

It is also recommended that we continue using mixed methods, combining quantitative and qualitative approaches to obtain a more comprehensive understanding of the subjective experiences of individuals diagnosed with BPD. In particular, future research should delve deeper into the impact of covert social violence and how gender stereotypes continue to influence the diagnoses and treatments of BPD.

## 5. Conclusions

This study has highlighted the connection between borderline personality disorder (BPD) symptoms, measured with the BSL-23, and experiences of covert social violence, assessed through the IVISEM scale. The BSL-23 results show high scores for emotions like shame and vulnerability, reflecting both BPD symptoms and the impact of the structural violence women face. The participants’ responses included terms such as “hormonal”, “period”, or “menstruation”, which reinforce the tendency to explain women’s emotional intensity biologically, rather than considering the social and structural factors that influence their emotional responses.

This study highlights the need for psychological interventions that address both BPD symptoms and the impact of covert and structural violence on women’s emotional health. The high scores on the BSL-23 and IVISEM suggest that interventions should consider both the emotional intensity linked to BPD and the social factors contributing to women’s emotional responses. It is crucial that we understand these emotional reactions within the context of social pressures and structural violence, avoiding their pathologization as an individual disorder.

The qualitative results highlight how women diagnosed with BPD experience the pathologization of their emotionality through gender stereotypes. Terms like “crazy” and “intense” often appeared in the responses, showing how women’s emotions are invalidated and can be misinterpreted as symptoms of BPD. This stigmatization reflects the gender stereotype of the emotionally overwhelmed woman. The roles imposed on women often conflict with their lived experiences, generating tensions between the assigned gender and the one performed [32]. These contradictions can lead women to suppress or integrate these roles to avoid being labeled as “bad”, “crazy”, or “sick” [31]. Quantitative results from the BSL-23 and IVISEM support these findings, showing high scores for emotional intensity and the impact of covert social violence, suggesting that these social pressures exacerbate the pathologization of emotions.

A large number of participants reported having experienced physical and sexual violence, which invites the reflection that BPD cannot be seen only as an individual disorder, but rather as a condition influenced by social and structural factors. This is supported by high scores on the IVISEM scale, which measures the impact of covert social violence. Furthermore, recurring themes related to social pressures, such as beauty standards and traditional gender roles, suggest that these external factors could exacerbate BPD symptoms, reinforcing the need for interventions that consider both the psychological and social dimensions of the disorder.

These findings underscore the importance of addressing structural and interpersonal violence in both research and clinical settings. Therapeutic approaches should go beyond individual pathology to incorporate an understanding of the broader social and systemic factors that shape these experiences. By doing so, interventions can better address the root causes of distress and support long-term recovery for those affected by BPD. Incorporating a feminist and systemic perspective into therapeutic practices is essential in order to ensure that interventions not only alleviate symptoms but also challenge the societal norms that perpetuate violence and inequality.

Clinically, this study highlights the need to rethink diagnostic and therapeutic frameworks using a biopsychosocial approach that integrates a critical gender perspective. Mental health professionals must adopt inclusive and unbiased tools that consider how social structures and gender norms influence mental health outcomes. Movements from feminist and academic perspectives have challenged the tendency to pathologize natural biological processes or normal reactions to oppressive or violent circumstances [7]. Beyond revising the diagnostic criteria, it is essential that we question the sociocultural contexts in which these criteria are applied, to prevent the reinforcement of stigma and inequality. Moreover, incorporating individuals diagnosed with BPD as active collaborators in shaping mental health practices and policies can offer deeper insights into the structural and contextual factors influencing their emotional well-being, challenging traditional hierarchies between patients and professionals.

Based on the findings, we propose several practical recommendations to address the gender bias in diagnostic practices. First, diagnostic tools (not only those used for diagnosis but also those developed as diagnostic frameworks), such as the DSM, should be revised to incorporate a broader sociocultural framework that considers the structural and systemic factors influencing mental health, particularly in women and from women’s perspectives. These revisions should also involve the mental health community in these processes. This could include an explicit questioning of the gendered assumptions underlying specific diagnostic criteria for disorders such as BPD. Second, we recommend implementing mandatory training programs for mental health professionals, not limited to those involved in diagnosis. These programs should focus on recognizing and mitigating gender biases in diagnostic and therapeutic practices while also studying and analyzing biases present in interventions themselves. These programs should integrate feminist and intersectional perspectives and actively involve the populations they aim to address. Individuals with BPD should be viewed not as subjects of study but as contributors to knowledge, enriching the understanding of gender stereotypes and structural violence and equipping professionals with tools with which to address these issues effectively. Finally, we propose developing educational initiatives aimed at raising awareness of covert social violence and its psychological consequences. By integrating these concepts into clinical training and practice, mental health professionals can adopt more inclusive and equitable approaches that validate and empower women’s experiences rather than pathologizing them.

The findings of this study have significant implications not only for clinical practice but also for policy development and societal awareness. In clinical practice, the results highlight the urgent need for mental health professionals to critically examine their diagnostic and therapeutic approaches. It is essential that we ensure that the diagnostic criteria, perspectives, and even certain categories are free from gender bias. This requires a broader perspective on what we are “psychologizing”: whether these are normal responses to abnormal situations, the origins of certain diagnoses, and how our own personal experiences and historical contexts influence our practices. A profound “rethinking” of psychology is necessary in order to ensure that we are alleviating suffering rather than adding to it. Are we reproducing structural violence in our practices?

We must promote research and therapeutic practice jointly, incorporating the voices of individuals who experience these conditions and diagnoses as active participants in therapeutic processes for others. The incorporation of training programs that integrate feminist and intersectional perspectives can help clinicians better understand the systemic factors influencing mental health, particularly in women, and adopt more equitable practices. If we truly embrace a biopsychosocial model, this approach must be reflected in how we address suffering, considering the patriarchal structural violence in which we live. It is crucial that we reflect on the foundations of psychology: who created it, for whom it was designed, and how we perpetuate hierarchical structures when addressing suffering. Those who experience mental distress are the best providers of knowledge about their conditions and life situations. It is essential that we review who benefits from our diagnostic practices and interventions. Who benefits from women being more assertive and less angry?

From a policy perspective, the study emphasizes the need to revise diagnostic frameworks, such as the DSM, to reflect a broader sociocultural context. There has been little progress in transcultural reviews of this manual, and few advances in gender perspectives since the 1980s. Policymakers must advocate for the inclusion of diverse gender perspectives in the development of diagnostic criteria and promote funding for research exploring the intersection of gender and mental health. Moreover, it is crucial that we analyze whether we are perpetuating forms of structural violence and seek more humane and just ways to support and alleviate mental health conditions.

At a societal level, these findings call for greater awareness of the role of covert social violence in shaping mental health outcomes. Public campaigns, social justice, and a more community-centered and less individualistic approach to mental health are essential in order to interpret these struggles collectively rather than individually. Educational initiatives can also help reduce stigma and challenge social norms that pathologize traits traditionally associated with women, such as emotional intensity. Addressing these issues collectively can contribute to the creation of more inclusive and effective mental health systems.

## Figures and Tables

**Figure 1 ijerph-22-00196-f001:**
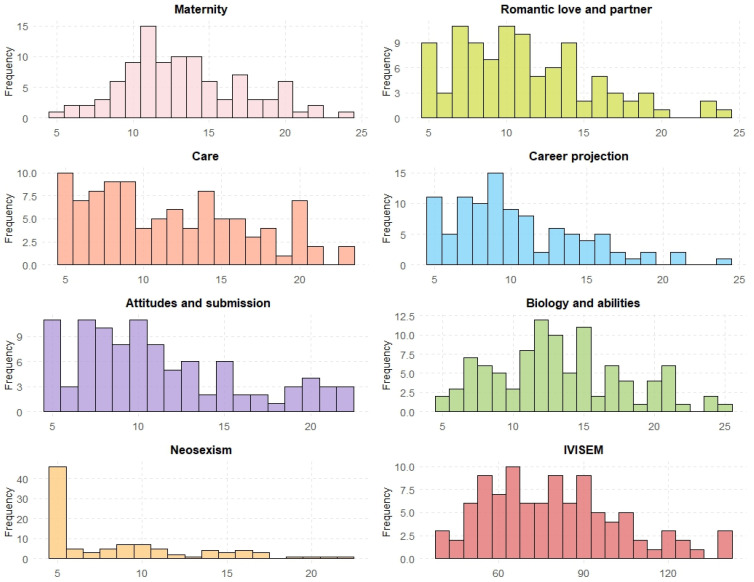
Frequencies of IVISEM subscales.

**Figure 2 ijerph-22-00196-f002:**
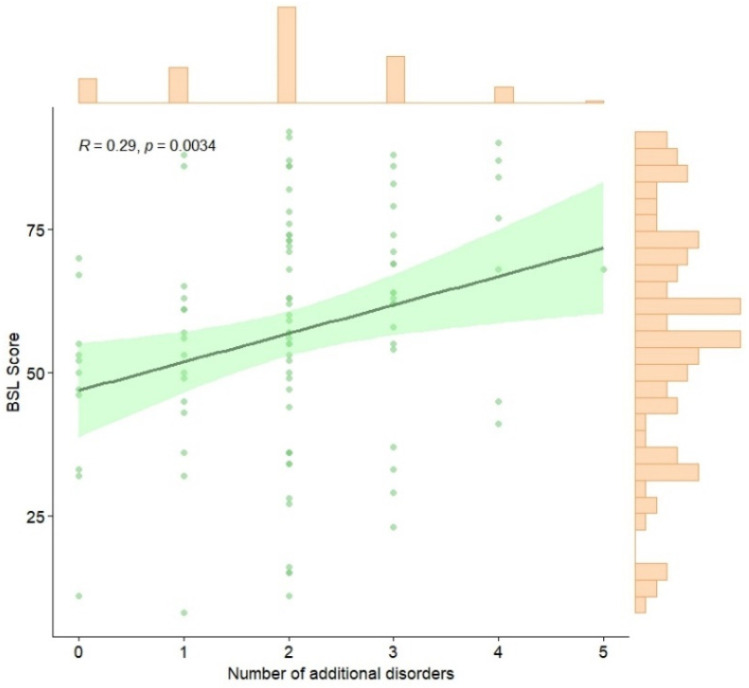
Spearman correlation between BSL total score and number of psychological disorders apart from BPD. Note: The scatterplot displays a positive trend between the total BSL score and the number of self-reported disorders, with a confidence interval represented by the green shaded area. Histograms along the axes illustrate the distributions of both variables. *R* = Spearman correlation coefficient.

**Table 1 ijerph-22-00196-t001:** Descriptives of the sample.

Variables	N = 99 ^1^
Gender	
Cisgender woman	88 (88.89%)
Cisgender man	5 (5.05%)
Non-binary	5 (5.05%)
Transgender woman	0 (0%)
Transgender man	1 (1.01%)
Other	0 (0%)
Perceived as a racialized person	
Yes	23 (23.23%)
No	76 (76.77%)
Nationality	
Spanish	43 (43.43%)
Other	56 (56.57%)
Civil status	
Single without a stable partner	43 (31.16%)
Single with a stable partner not cohabiting	25 (18.12%)
Single cohabiting with a stable partner	15 (10.87%)
Married	12 (8.7%)
Separated/Divorced	43 (31.16%)
Widowed	0 (0%)
Educational level achieved	
No education	0 (0%)
Primary education	4 (4.04%)
Secondary education	48 (48.48%)
University education	47 (47.47%)
Employment status	
Employed (working for others)	33 (33.33%)
Self-employed	7 (7.07%)
Working online	1 (1.01%)
Unemployed	21 (21.21%)
Retired	2 (2.02%)
Studying and working	19 (19.19%)
Other	16 (16.16%)
Economic status	
Comfortable, I can afford some luxuries	33 (53.23%)
I don’t have major problems, but I can’t afford luxuries	7 (11.29%)
It’s hard to make ends meet	1 (1.61%)
I have serious financial problems	21 (33.87%)
Sexual orientation	
Heterosexual	30 (30.3%)
Lesbian	29 (29.29%)
Bisexual	25 (25.25%)
Asexual	15 (15.15%)
Gay	0 (0%)
Prefer not to say	0 (0%)
Additional psychological disorders (besides BPD)	
0	11 (11.11%)
1	16 (16.16%)
2	43 (43.43%)
3	21 (21.21%)
4	7 (7.07%)
5	1 (1.01%)

^1^ *n* (%).

**Table 2 ijerph-22-00196-t002:** Statistical information regarding the BSL.

Item	*n*	Mean	SD	Median	MAD	Min	Max	Skewness	Kurtosis
BSL1	99	3.00	1.03	3	1.48	0	4	−0.72	−0.45
BSL2	99	2.72	1.24	3	1.48	0	4	−0.74	−0.52
BSL3	99	2.29	1.39	2	1.48	0	4	−0.25	−1.20
BSL4	99	2.14	1.44	2	1.48	0	4	−0.08	−1.34
BSL5	99	2.20	1.50	2	1.48	0	4	−0.24	−1.40
BSL6	99	2.94	1.01	3	1.48	0	4	−0.88	0.40
BSL7	99	1.83	1.58	2	2.97	0	4	0.13	−1.58
BSL8	99	3.13	1.16	4	0.00	0	4	−1.27	0.68
BSL9	99	3.32	1.00	4	0.00	0	4	−1.77	2.93
BSL10	99	2.49	1.34	3	1.48	0	4	−0.56	−0.81
BSL11	98	2.58	1.47	3	1.48	0	4	−0.53	−1.14
BSL12	99	2.01	1.56	2	2.97	0	4	−0.02	−1.53
BSL13	99	2.69	1.31	3	1.48	0	4	−0.70	−0.72
BSL14	99	3.06	1.25	4	0.00	0	4	−1.01	−0.26
BSL15	99	1.17	1.41	1	1.48	0	4	0.81	−0.78
BSL16	99	2.49	1.45	3	1.48	0	4	−0.43	−1.23
BSL17	99	3.14	1.08	3	1.48	0	4	−1.24	0.92
BSL18	99	1.93	1.66	2	2.97	0	4	0.10	−1.65
BSL19	99	2.36	1.52	3	1.48	0	4	−0.40	−1.36
BSL20	99	2.46	1.39	3	1.48	0	4	−0.50	−1.03
BSL21	99	2.25	1.53	3	1.48	0	4	−0.27	−1.43
BSL22	99	2.01	1.56	2	2.97	0	4	−0.11	−1.55
BSL23	99	2.63	1.50	3	1.48	0	4	−0.63	−1.12
Total	99	56.81	20.31	57	19.27	8	92	−0.37	2.04

Note: *n* = sample size, SD = standard deviation, MAD = median absolute deviation, Min = minimum value, and Max = maximum value.

**Table 3 ijerph-22-00196-t003:** Statistical information regarding the IVISEM.

Item	*n*	Subscale	Mean	SD	Median	MAD	Min	Max	Skewness	Kurtosis
IVISEM1	99	Maternity	3.26	1.28	3	1.48	1	5	−0.35	−0.86
IVISEM2	99	Romantic Love and Partner	3.21	1.29	3	1.48	1	5	−0.39	−0.88
IVISEM3	99	Care	3.06	1.47	3	1.48	1	5	−0.14	−1.33
IVISEM4	99	Career Projection	1.76	1.24	1	0	1	5	1.52	1.05
IVISEM5	99	Submission Attitudes	2.05	1.23	1	0	1	5	0.75	−0.69
IVISEM6	99	Biology and Abilities	1.70	1.16	1	0	1	5	1.50	1.06
IVISEM7	99	Neosexism	1.85	1.27	1	0	1	5	1.16	−0.06
IVISEM8	99	Maternity	3.08	1.52	3	1.48	1	5	−0.13	−1.48
IVISEM9	99	Romantic Love and Partner	2.30	1.31	2	1.48	1	5	0.59	−0.85
IVISEM10	99	Care	2.29	1.48	2	1.48	1	5	0.66	−1.11
IVISEM11	99	Career Projection	1.53	0.96	1	0	1	5	1.80	2.51
IVISEM12	99	Submission Attitudes	2.51	1.45	2	1.48	1	5	0.32	−1.41
IVISEM13	99	Biology and Abilities	3.05	1.38	3	1.48	1	5	−0.14	−1.20
IVISEM14	99	Neosexism	1.39	0.85	1	0	1	5	2.17	4.00
IVISEM15	99	Maternity	2.07	1.33	1	0	1	5	0.92	−0.47
IVISEM16	99	Romantic Love and Partner	1.97	1.22	1	0	1	5	1.00	−0.13
IVISEM17	99	Care	2.20	1.44	1	0	1	5	0.68	−1.14
IVISEM18	99	Career Projection	2.45	1.48	2	1.48	1	5	0.43	−1.33
IVISEM19	99	Submission Attitudes	2.23	1.43	2	1.48	1	5	0.66	−1.07
IVISEM20	99	Biology and Abilities	2.92	1.40	3	1.48	1	5	0.01	−1.33
IVISEM21	99	Neosexism	1.77	1.27	1	0	1	5	1.42	0.58
IVISEM22	99	Maternity	2.99	1.37	3	1.48	1	5	−0.08	−1.19
IVISEM23	99	Romantic Love and Partner	2.16	1.21	2	1.48	1	5	0.68	−0.58
IVISEM24	99	Care	2.18	1.32	2	1.48	1	5	0.62	−1.04
IVISEM25	99	Career Projection	2.90	1.42	3	1.48	1	5	−0.14	−1.43
IVISEM26	99	Submission Attitudes	2.67	1.44	3	1.48	1	5	0.12	−1.39
IVISEM27	99	Biology and Abilities	2.96	1.46	3	1.48	1	5	−0.01	−1.41
IVISEM28	99	Neosexism	1.93	1.25	1	0	1	5	1.04	−0.19
IVISEM29	99	Maternity	1.92	1.20	1	0	1	5	0.96	−0.39
IVISEM30	99	Romantic Love and Partner	1.49	0.90	1	0	1	5	1.99	3.76
IVISEM31	99	Care	1.95	1.27	1	0	1	5	1.09	−0.10
IVISEM32	99	Career Projection	1.70	1.22	1	0	1	5	1.58	1.16
IVISEM33	99	Submission Attitudes	1.72	1.16	1	0	1	5	1.41	0.80
IVISEM34	99	Biology and Abilities	2.74	1.49	3	1.48	1	5	0.17	−1.39
IVISEM35	99	Neosexism	1.59	1.11	1	0	1	5	1.93	2.87
Maternity	99	-	13.32	3.97	13	2.97	5	24	0.44	−0.32
Romantic Love and Partner	99	-	11.14	4.40	10	4.45	5	24	0.73	0.12
Care	99	-	11.69	5.01	11	5.93	5	23	0.43	−0.93
Career Projection	99	-	10.33	4.15	9	2.97	5	24	0.90	0.40
Submission Attitudes	99	-	11.17	4.81	10	4.45	5	22	0.72	−0.47
Biology and Abilities	99	-	13.36	4.70	13	4.45	5	25	0.36	−0.52
Neosexism	99	-	8.53	4.48	6	1.48	5	22	1.14	0.30
Total	99	-	79.55	23.68	78	23.72	41	142	0.63	−0.13

Note: *n* = sample size, SD = standard deviation, MAD = median absolute deviation, Min = minimum value, and Max = maximum value.

**Table 4 ijerph-22-00196-t004:** Summary of thematic categories and most frequent keywords.

Thematic Category	Most Frequent Keywords	Count	Weighted Percentage
Stigmatization and Violence	Woman, crazy, women	372, 273, 229	1.24%, 0.91%, 0.77%
Anger and Intense Emotions	Emotional, Anger, dramatic	133, 74, 73	0.44%, 0.25%, 0.24%
Trauma and Attachment	Abuse, sexual, guilt	5, 4, 2	1.07%, 0.86%, 0.43%
Opinion on the BPD Label	Diagnosis, labeled, treatment	101, 68, 10	0.34%, 0.23%, 0.41%
Structural Violence	Gender norms, emotional expression	50, 42, 35	0.16%, 0.14%, 0.12%
The ‘Crazy’ Label as a Tool	Invalidation, minimization, gender bias	28, 25, 20	0.12%, 0.10%, 0.08%
Cultural and Gender Norms	Social norms, cultural values, expectations	33, 28, 22	0.13%, 0.11%, 0.09%
TOTAL	Woman, crazy, women, period (“regla” and “periodo”) exaggerating, emotional, hysterical, sensible, hormonal, aggressive, dramatics, intense	372, 273, 229, 141, 134, 133, 106, 103, 87, 77, 73, 64	1.24%, 0.91%, 0.77%, 0.47%, 0.45%, 0.44%, 0.35%, 0.34%, 0.29%, 0.26%, 0.24%, 0.21%

## Data Availability

The data supporting the results of this study are not publicly available due to ethical and privacy restrictions. Access to the data may be requested from the corresponding author under reasonable conditions, provided appropriate ethical clearance is obtained.

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
