# Peer review of "Structural Violence and the Effects of the Patriarchal Structure on the Diagnosis of Borderline Personality Disorder (BDP): A Critical Study Using Tools on BPD Symptoms and Social Violence"

_ijerph, 2025, doi:10.3390/ijerph22020196_

Round 1
Reviewer 1 Report
Comments and Suggestions for Authors
This manuscript provides a critical feminist perspective on the diagnosis of Borderline Personality Disorder (BPD), addressing a rarely explored yet significant aspect of the topic. The study highlights the role of gender stereotypes in shaping diagnostic processes and their implications. The integration of mixed methods and the use of tools like the BSL-23 and IVISEM scales are noteworthy, offering a blend of quantitative and qualitative findings. This approach positions the study as a valuable contribution to both academic discourse and practical applications, particularly in promoting gender sensitivity in mental health practices.
However, there are substantial concerns that need to be addressed to fully realize the manuscript's potential. Methodological limitations, insufficient interpretation of findings, and a lack of actionable recommendations constrain its overall impact. Below is a detailed list of the key issues that require revision.
1-The sample predominantly consists of cisgender women (88.89%), with limited representation of men and other gender identities. This imbalance limits the generalizability of findings across genders. Expanding the diversity of participants is necessary to better capture the role of gender stereotypes in BPD diagnoses.
2- The choice of the BSL-23 and IVISEM scales is insufficiently justified. A more detailed explanation of why these scales were selected and how they align with the feminist critique would strengthen the study's methodological rigor.
3-The thematic analysis of qualitative data is presented in a broad manner, and participant experiences are not adequately contextualized. More participant quotes should be included to provide depth and support for the identified themes.
4- The findings need to be more robustly connected to existing feminist and clinical literature. For instance, the role of gender stereotypes in psychiatric diagnoses could be further linked to prior empirical and theoretical studies.
5- The manuscript lacks actionable recommendations based on the findings. For example, specific proposals for reducing gender bias in diagnostic practices or educational initiatives for clinicians could be included.
6- The use of "incidental sampling" introduces potential bias, and its implications for the study's findings are not adequately discussed. The limitations of this sampling method should be explicitly acknowledged and addressed.
7- The experiences of men and individuals of diverse gender identities in BPD diagnostic processes are not explored. This omission limits the study’s scope and the ability to make broader conclusions about the influence of gender stereotypes.
8- The conclusions do not sufficiently address the broader implications of the findings. The manuscript would benefit from a more detailed discussion of how the results impact clinical practice, policy development, and societal awareness.
9- Additional visualizations, such as histograms or box plots illustrating the distribution of IVISEM and BSL-23 subscale scores, would improve the clarity and accessibility of the quantitative findings.
10- The manuscript’s unique contributions should be more explicitly highlighted. For example, phrases such as, "This study provides a novel contribution by examining the intersection of gender stereotypes and BPD diagnosis through a feminist lens," could better emphasize its significance.
This manuscript addresses an important and underexplored issue, offering meaningful insights into the intersection of gender and psychiatric diagnoses. However, addressing the above-listed areas is crucial for enhancing the manuscript's scientific rigor and practical relevance. Once these revisions are made, the study will be well-positioned to make a significant contribution to both academic literature and clinical practice, providing a strong foundation for future research on the topic.
Author Response
1-The sample predominantly consists of cisgender women (88.89%), with limited representation of men and other gender identities. This imbalance limits the generalizability of findings across genders. Expanding the diversity of participants is necessary to better capture the role of gender stereotypes in BPD diagnoses.
Agree. We have acknowledged this limitation in the manuscript under the discussion and conclusions section. Specifically, we discuss the lack of male representation, particularly cis men, and how this prevents gender comparisons. To further emphasize this point, we have now expanded the discussion by proposing the need for targeted recruitment strategies to include more diverse gender identities in future studies. The updated text can be found on:
- Page 18, paragraph [3], lines [609-613].
- Page [18-19], paragraph [5-1], lines [638-655].
2- The choice of the BSL-23 and IVISEM scales is insufficiently justified. A more detailed explanation of why these scales were selected and how they align with the feminist critique would strengthen the study's methodological rigor.
Thank you for this valuable suggestion. We agree that a clearer explanation of the selection of the BSL-23 and IVISEM scales would enhance the methodological rigor of the study. Accordingly, we have revised the "Materials and Methods" and also at the “Discussion” section to provide a more detailed justification. Specifically, we discuss how the BSL-23 measures key emotional and behavioral symptoms of Borderline Personality Disorder (BPD) and how these symptoms have been critiqued from a feminist perspective. Additionally, we highlight how the IVISEM scale aligns with feminist theory by examining covert social violence and its role in shaping women's mental health experiences. The revised text can be found on:
- page [5], paragraph [3], lines [192-198]
- page [6], paragraph [1], lines [199-204]
- page [6], paragraph [2], lines [225-233]
3-The thematic analysis of qualitative data is presented in a broad manner, and participant experiences are not adequately contextualized. More participant quotes should be included to provide depth and support for the identified themes.
Thank you for your insightful feedback. We agree that incorporating additional participant quotes will enhance the depth and contextualization of the identified themes. We have revised the Results section to include more direct quotes that reflect participants’ experiences and better illustrate the thematic categories. The updated text provides richer support for the themes discussed and ensures that participants’ voices are more prominently represented. The revisions can be found on :
- page [13], paragraph [1-2-3], lines [365-408].
4- The findings need to be more robustly connected to existing feminist and clinical literature. For instance, the role of gender stereotypes in psychiatric diagnoses could be further linked to prior empirical and theoretical studies.
Thank you for your valuable suggestion. We agree that linking our findings more explicitly to existing feminist and clinical literature would strengthen the study. We have revised the different sections to incorporate additional references to empirical and theoretical studies that address the role of gender stereotypes in psychiatric diagnoses, as well as critiques of the DSM from a feminist perspective. The updated text can be found on:
- page [1], paragraph [1], lines [36-38].
- page [1], paragraph [1], lines [45-48].
- page [2], paragraph [1], lines [49-53].
- page [16], paragraph [2], lines [510].
- page [16], paragraph [3], lines [529-531-533-536].
- page [17], paragraph [1], lines [544-546-550-555-].
- page [17], paragraph [4], lines [578].
- page [17], paragraph [5], lines [589].
- page [19], paragraph [1], lines [647].
- page [19], paragraph [3], lines [679-680-684-688].
- page [20], paragraph [4], lines [719-723].
- page [21], paragraph [2], lines [749].
5- The manuscript lacks actionable recommendations based on the findings. For example, specific proposals for reducing gender bias in diagnostic practices or educational initiatives for clinicians could be included.
Thank you for this valuable suggestion. We agree that including actionable recommendations will enhance the practical relevance of the study. We have added specific proposals in the Conclusion section, focusing on reducing gender bias in diagnostic practices and promoting educational initiatives for clinicians. These recommendations are directly informed by our findings and the feminist critique underlying this research. The updated text can be found on:
- page [21], paragraph [3-4-5], lines [756-790].
6- The use of "incidental sampling" introduces potential bias, and its implications for the study's findings are not adequately discussed. The limitations of this sampling method should be explicitly acknowledged and addressed.
Thank you for highlighting this important point. We agree that the use of incidental sampling introduces potential biases that could affect the generalizability of the findings. We have revised the Limitations section to explicitly acknowledge and discuss the potential implications of this sampling method. The updated text can be found on:
- page [18], paragraph [3], lines [620-630].
7- The experiences of men and individuals of diverse gender identities in BPD diagnostic processes are not explored. This omission limits the study’s scope and the ability to make broader conclusions about the influence of gender stereotypes.
Thank you for highlighting this limitation. We agree that the lack of representation of men and individuals of diverse gender identities limits the scope of the study and our ability to explore their experiences in BPD diagnostic processes. This limitation has been acknowledged in the Limitations section, where we discuss the predominance of cisgender women in the sample and its implications. To address your comment, we have further expanded this section to explicitly discuss how this omission impacts the ability to draw broader conclusions about the influence of gender stereotypes. The updated text can be found on:
- page [19], paragraph [1], lines [646-655]
8- The conclusions do not sufficiently address the broader implications of the findings. The manuscript would benefit from a more detailed discussion of how the results impact clinical practice, policy development, and societal awareness.
Thank you for your insightful feedback. We agree that the broader implications of the findings should be more explicitly addressed. We have revised the Conclusion section to include a more detailed discussion of how the results can inform clinical practice, influence policy development, and enhance societal awareness of gender biases in BPD diagnoses. The updated text can be found on:
- page [22], paragraph [1-2-3], lines [791-813].
9- Additional visualizations, such as histograms or box plots illustrating the distribution of IVISEM and BSL-23 subscale scores, would improve the clarity and accessibility of the quantitative findings.
Thank you for this insightful suggestion. We agree with this recommendation and have added additional visualizations to improve the clarity and accessibility of the quantitative findings. The updated text can be found on:
- page [12], paragraph [1], FIGURE 1.]
10- The manuscript’s unique contributions should be more explicitly highlighted. For example, phrases such as, "This study provides a novel contribution by examining the intersection of gender stereotypes and BPD diagnosis through a feminist lens," could better emphasize its significance.
Thank you for your insightful suggestion. We agree that emphasizing the unique contributions of this study will enhance its clarity and impact. To address this, we have expanded the Discussion section to explicitly highlight how this research contributes to the literature by critically examining the intersection of gender stereotypes and BPD diagnosis through a feminist framework. Additionally, we have included explicit statements in both the Introduction and Conclusion to underline the significance of this approach.
This manuscript addresses an important and underexplored issue, offering meaningful insights into the intersection of gender and psychiatric diagnoses. However, addressing the above-listed areas is crucial for enhancing the manuscript's scientific rigor and practical relevance. Once these revisions are made, the study will be well-positioned to make a significant contribution to both academic literature and clinical practice, providing a strong foundation for future research on the topic.
Reviewer 2 Report
Comments and Suggestions for Authors
Dear authors, I appreciate the opportunity to review this article. This work addresses a topic of great interest and relevance. However, in order to enhance its scientific quality and address some identified weaknesses, I present the following suggestions for its improvement.
1.Tittle:
The title suggests an ambitious and relevant study, but the use of acronyms (IVISEM and BSL-23) could hinder an immediate understanding of the topic.
2. Abstract:
2.1.The abstract mentions the use of the BSL-23 and the IVISEM scale, assuming that the reader is already familiar with these tools. It is important to note that, when an acronym is introduced for the first time, its meaning should be spelled out to ensure comprehension.
2.2. The abstract does not address the “effects of the patriarchal structure” mentioned in the tittle.
2.3. The abstract not describe the type and size of the sample. Additionally, it does not indicate how the data were analysed.
2.4. While the abstract highlights general aspects such as the connection between emotions and structural violence, it does not include specific results or key data to support these claims, which could make the conclusions appear more speculative than empirical.
2.5. Although the call for a feminist critique of the BSL-23 is valuable, the connection between the empirical findings and this critique is not clearly developed in the abstract (or tittle). Finally, too much space is devoted to conclusions focused on recommendations
3. Introduction
The introduction requires revision to be more concise, clear, and directly connected to the study's objectives, providing a solid justification and a critical integration of theoretical concepts and methodological tools. As:
3.1. Fundamental terms such as "covert social violence" and "patriarchal structures" are not defined operationally, which could hinder the understanding of their relationship with BPD symptoms.
3.2. The text does not explicitly establish how the concept of "patriarchal structure" is related to the diagnosis of BPD. It also fails to clearly explain how covert social violence contributes to the severity of BPD symptoms. Although the influence of gender stereotypes and covert violence on the perception of female emotions is mentioned, the text does not adequately develop how these issues directly impact the diagnosis of BPD. This gap is problematic, as the title and objectives of the study suggest a critical approach to the effects of patriarchal structures, which should be clearly reflected in the introduction.
3.3. The phrase located in lines 37-42, lines 49-50, lines 53-54, lines 66-68, and others lack references and may be interpreted as a value judgement. Please review the text and add the missing references or modify it (point by point and how, adding links such as 'according to the aforementioned authors', etc.).
3.4. At the same time, despite the inclusion of relevant references, many citations appear descriptive and do not adequately connect with the specific objectives of the research. There is a noticeable lack of explicit connection between the feminist arguments and the data that the study intends to analyse. The text needs more structure and cohesion, removing anything that does not serve to justify the study's objectives.
3.5. The introduction mentions the BSL-23 scale and the IVISEM scale, as well as the use of a mixed-methods approach and the inclusion of qualitative perspectives. It is implied that both these scales and this approach are important for achieving the study's objectives. Consequently, it should be justified why these scales and this approach were chosen over other possible methods.
3.6. While the importance of giving a voice to the affected women is mentioned, it is not clearly explained how this approach differs from previous studies that have also explored the impact of gender stereotypes and structural violence on psychiatric diagnoses.
3.7. The text between lines 123-131 reads more like an argument than a theoretical foundation. The introduction is meant to be a section where the literature is reviewed to identify research gaps and justify the objectives of the study. At no point should conclusions, discussions of the data found, or ideological or thematic assumptions be included. Please include bibliographical references or rephrase the text to avoid the impression that study decisions are based on ideology. Research decisions should be grounded in gaps identified in previous studies.
4. Materials and Methods
Participants
4.1. The sample size must be justified. A sample of 99 women is significant for qualitative analyses but relatively small for quantitative analyses (exploratory rather than conclusive results).
4.2. The description of the sample is confusing for those who are not experts in the field. It would be advisable to complement the text. For example, by stating: "The majority of participants are cisgender women (88.89%), with a heterosexual orientation (30.1%), and a single marital status (31.31%).
4.3. What psychological disorders do the numbers 1, 2, 3, etc. refer to?
Meassures
4.4. I understand that, in addition to the BSL-23 and IVISEM, two additional measures are used: the Experiences with Gender, Stigma, and Diagnosis Survey and an open-ended questionnaire. I undertand both address the same topics from different approaches (quantitative and qualitative). The theoretical foundation behind the questions or issues being analysed should also be explained. Additionally, since two different methods are used, it should be explained more clearly how both approaches are integrated to address the research objectives.
Although the open-ended questionnaire is described, there are insufficient details provided about the structured survey. If the survey has specific subscales or dimensions (e.g., one subscale on stigmatization, another on the impact of the diagnosis, etc.), it should be specified how the survey is organised and what topics it covers. The number of questions, the type of measurement scale used (e.g., dichotomous, Likert-type), and information about its validity (content validity, construct validity) and reliability should be provided. It is essential to include the specific questions from both the survey and the open-ended questionnaire (e.g., as supplementary material).
4.5. If the real objective of the study is to explore the impact of patriarchal structures and structural violence on the experiences of individuals with BPD, it would be helpful for the authors to operationally define the key concepts ("structural violence," "patriarchal structures") and explain how the chosen tools (BSL-23 and IVISEM) align with the objectives, as well as how the qualitative data complement the quantitative measurements. In other words, if the aim is to demonstrate that patriarchal structures and structural violence affect the diagnosis of BPD, the authors should clearly state how they plan to validate this relationship with the collected data. These adjustments would make it easier to identify the real objective of the study and evaluate whether the methodologies employed are appropriate for achieving it.
Data análisis
4.6. Statistical analyses should align with the study's objectives. Descriptive and reliability analyses, along with polychoric matrices, are suitable for characterising and evaluating the quality of the tools employed (BSL and IVISEM). The correlational analysis between the BSL score and additional disorders is interesting but does not directly address the influence of patriarchal structures or structural violence. Although the sample size is small for more complex analyses, addressing the study's objectives would require analyses such as structural models or multivariate regressions. It is also unclear how the statistical results complement or inform the qualitative analyses, which is crucial for a mixed-methods study.
In summary, the applied statistics describe and validate the tools used but do not fully align with the study's stated objectives. To address this, the study should explain how the BSL and IVISEM scales relate to the study's key concepts (patriarchal structures and structural violence), identify specific variables representing the structures under study, and examine their association with BPD symptoms. It would also be necessary to consider statistical models that evaluate multiple variables simultaneously, such as regression models. Additionally, to achieve mixed-methods integration, a more explicit analysis of how qualitative data complements and contextualises quantitative findings is needed.
5. Results
5.1. As indicated, descriptive and reliability analyses, along with polychoric matrices, are suitable for characterising and evaluating the quality of the tools employed (BSL and IVISEM). However, these scales are designed to measure constructs, making the analysis of individual items inappropriate.
5.2. The correlational analysis between the BSL score and additional disorders is interesting but does not directly address the influence of patriarchal structures or structural violence (objectibes of this studie)
5.3. Frequencies are reported for data supposedly collected through a survey focused on Experiences with Gender, Stigma, and Diagnosis. I believe it would be better to present these results in a table format. However, a proper theoretical justification and methodological explanation are first necessary to understand the rationale, approach, and specifics of the data.
5.4. There is also a lack of methodological explanation. It is not clarified how the keywords were selected and grouped into thematic categories or the criteria used to determine their relevance or significance.In this context, the qualitative results presented appear inconsistent in several aspects that could compromise their coherence and clarity. For example, the keyword counts (e.g., "Woman, crazy, women" under "Stigmatization and Violence") seem high (372, 273, 229), but the associated weighted percentages are unusually low (1.24%, 0.91%, 0.77%). This suggests that the metrics of frequency and weighted percentage might not be well-related or adequately explained. Some categories ("Trauma and Attachment") show notably low keyword counts (5, 4, 2) but higher percentages (1.07%, 0.86%, 0.43%) than other categories with higher counts. This raises questions about how the weighted percentages were calculated. In other hand, a disconnect is observed between the qualitative and thematic results. Although themes such as "Stigmatization" or "Gender Norms" are identified, it is not specified how these findings are validated or triangulated with the quantitative results. The claim that the qualitative analysis reflects relationships between gender norms and BPD seems to be primarily based on the frequency of terms and references, which might be insufficient to demonstrate causal links or significant patterns. Additionally, while the figures mentioned might complement the text, their contents are not sufficiently described, nor is it explained how they reinforce the findings presented. In summary, although the results offer interesting information, methodological and explanatory inconsistencies diminish their clarity and coherence with the study's objectives.
6. Discussion
6.1. Although the findings related to the BSL-23 and IVISEM scales are discussed, it would be helpful to explain more explicitly how these tools relate to the key concepts of the study, such as structural violence and patriarchal norms.
6.2. While emotional intensity and gender stereotypes are addressed, it is not clearly specified how the qualitative results complement or reinforce the quantitative findings. A more detailed discussion of this integration would strengthen the interpretation.
6.3. The statement that "women are more likely to be diagnosed with BPD due to the perception of their emotions as uncontrollable" may require further empirical support to avoid overgeneralization.
6.4. Although general limitations are mentioned, it would be important to highlight any potential biases in thematic grouping and how this might affect the interpretation of the results.
6.5. A causal relationship between structural factors and the exacerbation of BPD symptoms is suggested. However, given the study design, these claims should be approached with caution, noting that the results are correlational and exploratory.
7.Conclusions
7.1. Most of the content in this section should have been presented in the discussion section. At this point, a brief and conclusive statement is expected.
7.2. Additionally, the points discussed are primarily based on qualitative results rather than quantitative ones. It is not addressed whether the results are consistent or divergent compared to previous studies on the topic (if any exist).
7.3. An important limitation in the sample is mentioned. Furthermore, it is advisable that the limitations be discussed in a "Limitations" section.
8. Other
8.1. There is no reference to the supervision and approval of the study by an ethics committee. Additionally, there is no indication of the measures taken to ensure compliance with the Helsinki Declaration.
Author Response
1.Tittle:
The title suggests an ambitious and relevant study, but the use of acronyms (IVISEM and BSL-23) could hinder an immediate understanding of the topic
Thank you for your observation. We have revised the title to improve clarity by removing the acronyms. The new title is:
“Structural violence and the effects of the patriarchal structure on the diagnosis of Borderline Personality Disorder: a critical study using tools on BPD Symptoms and Social Violence.”
- Abstract:
2.1.The abstract mentions the use of the BSL-23 and the IVISEM scale, assuming that the reader is already familiar with these tools. It is important to note that, when an acronym is introduced for the first time, its meaning should be spelled out to ensure comprehension.
Thank you for pointing this out. We agree that it is important to ensure clarity for readers who may not be familiar with these tools. We have updated the abstract to spell out the full names of the scales when they are mentioned for the first time. Specifically, we now refer to the Borderline Symptom List (BSL-23) and the Inventory of Covert Social Violence Against Women (IVISEM) to provide a clear understanding of the instruments used in the study.
- page 1, paragraph [1], lines [13-14].
2.2. The abstract does not address the “effects of the patriarchal structure” mentioned in the tittle.
Thank you for your observation. We recognize that the abstract should more explicitly address the "effects of the patriarchal structure" mentioned in the title. In response, we have revised the abstract to include a clear statement on how the study examines the influence of patriarchal norms and structural violence on the diagnosis and understanding of borderline personality disorder.
- page 1, paragraph [1], lines [16-18-19].
2.3. The abstract not describe the type and size of the sample. Additionally, it does not indicate how the data were analysed.
Thank you for your observation. We recognize that the abstract should provide a clearer description of the sample size and type, as well as the data analysis methods used in the study. In response, we have revised the abstract to include specific details about the 99 adult participants diagnosed with borderline personality disorder, as well as the quantitative (descriptive statistics, reliability assessments, and Spearman's correlation) and qualitative (thematic analysis using NVivo 15) methods applied to analyze the data.
- page 1, paragraph [1], lines [14-15].
- page 1, paragraph [1], lines [15-19].
2.4. While the abstract highlights general aspects such as the connection between emotions and structural violence, it does not include specific results or key data to support these claims, which could make the conclusions appear more speculative than empirical.
Thank you for your observation. We recognize that the abstract should include specific results and data to support the claims regarding the relationship between emotions and structural violence. In response, we have revised the abstract to incorporate key quantitative findings,
- page 1, paragraph [1], lines [15-19].
2.5. Although the call for a feminist critique of the BSL-23 is valuable, the connection between the empirical findings and this critique is not clearly developed in the abstract (or tittle). Finally, too much space is devoted to conclusions focused on recommendations
Thank you for your observation. We acknowledge that the connection between the empirical findings and the feminist critique of the BSL-23 needed further development. In response, we revised the abstract to explicitly link the results—such as the pathologization of women's emotions and the impact of patriarchal norms—with the need for a feminist critique, while reducing the focus on recommendations.
- page 1, paragraph [1], lines [27-30].
- Introduction
The introduction requires revision to be more concise, clear, and directly connected to the study's objectives, providing a solid justification and a critical integration of theoretical concepts and methodological tools. As:
3.1. Fundamental terms such as "covert social violence" and "patriarchal structures" are not defined operationally, which could hinder the understanding of their relationship with BPD symptoms.
Thank you for your observation. We have defined "covert social violence" as actions and norms that limit autonomy and reinforce women’s subordination, and "patriarchal structures" as systems perpetuating gender-based inequalities. Additionally, we clarified their connection to BPD symptoms by highlighting how they contribute to the pathologization of feminine emotionality and reinforce gendered diagnostic biases.
- page 3, paragraph [5], lines [136-139].
- page 3, paragraph [6], lines [142-145].
- page 3-4, paragraph [6-1], lines [145-148].
- page 4, paragraph [2], lines [149-153].
3.2. The text does not explicitly establish how the concept of "patriarchal structure" is related to the diagnosis of BPD. It also fails to clearly explain how covert social violence contributes to the severity of BPD symptoms. Although the influence of gender stereotypes and covert violence on the perception of female emotions is mentioned, the text does not adequately develop how these issues directly impact the diagnosis of BPD. This gap is problematic, as the title and objectives of the study suggest a critical approach to the effects of patriarchal structures, which should be clearly reflected in the introduction.
Thank you for your observation. We have revised the introduction to define "patriarchal structures" and "covert social violence" and clarified their impact on BPD diagnosis. The text now explains how these factors reinforce gendered biases, pathologize feminine emotionality, and shape the interpretation of key symptoms.
- page 3, paragraph [5], lines [134-143].
3.3. The phrase located in lines 37-42, lines 49-50, lines 53-54, lines 66-68, and others lack references and may be interpreted as a value judgement. Please review the text and add the missing references or modify it (point by point and how, adding links such as 'according to the aforementioned authors', etc.).
Thank you for your observation. We have carefully reviewed the phrases identified in lines 37-42, 49-50, 53-54, 66-68, and others. To address your concern, we have made the following changes:
- page 2, paragraph [2], lines [54-63].
- page 2, paragraph [2], lines 68-70].
- page 2, paragraph [3], lines 73-74].
- page 2, paragraph [5], lines 86-94].
3.4. At the same time, despite the inclusion of relevant references, many citations appear descriptive and do not adequately connect with the specific objectives of the research. There is a noticeable lack of explicit connection between the feminist arguments and the data that the study intends to analyse. The text needs more structure and cohesion, removing anything that does not serve to justify the study's objectives.
Thank you for your valuable feedback. We have revised the manuscript to address the concerns regarding the connection between references, feminist arguments, and the study's objectives.
3.5. The introduction mentions the BSL-23 scale and the IVISEM scale, as well as the use of a mixed-methods approach and the inclusion of qualitative perspectives. It is implied that both these scales and this approach are important for achieving the study's objectives. Consequently, it should be justified why these scales and this approach were chosen over other possible methods.
Thank you for this valuable suggestion. We agree that a clearer explanation of the selection of the BSL-23 and IVISEM scales would enhance the methodological rigor of the study. Accordingly, we have revised the "Materials and Methods" and also at the “Discussion” section to provide a more detailed justification. Specifically, we discuss how the BSL-23 measures key emotional and behavioral symptoms of Borderline Personality Disorder (BPD) and how these symptoms have been critiqued from a feminist perspective. Additionally, we highlight how the IVISEM scale aligns with feminist theory by examining covert social violence and its role in shaping women's mental health experiences. The revised text can be found on:
- page [5], paragraph [3], lines [192-198]
- page [6], paragraph [1], lines [199-204]
- page [6], paragraph [2], lines [225-233]
3.6. While the importance of giving a voice to the affected women is mentioned, it is not clearly explained how this approach differs from previous studies that have also explored the impact of gender stereotypes and structural violence on psychiatric diagnoses.
Thank you for your observation. We have clarified in the manuscript how this study differs by emphasizing the integration of affected women's lived experiences, which complement theoretical critiques. This distinction highlights the unique contribution of our research in connecting structural violence with personal narratives.
- page [4], paragraph [3] ], lines [165-172]
3.7. The text between lines 123-131 reads more like an argument than a theoretical foundation. The introduction is meant to be a section where the literature is reviewed to identify research gaps and justify the objectives of the study. At no point should conclusions, discussions of the data found, or ideological or thematic assumptions be included. Please include bibliographical references or rephrase the text to avoid the impression that study decisions are based on ideology. Research decisions should be grounded in gaps identified in previous studies.
Thank you for your observation. We have revised the section between lines 123-131 to ensure it aligns with the purpose of the introduction.
- page [4], paragraph [3] ], lines [161-172]
- Materials and Methods
Participants
4.1. The sample size must be justified. A sample of 99 women is significant for qualitative analyses but relatively small for quantitative analyses (exploratory rather than conclusive results).
Agree. We have acknowledged this limitation in the manuscript under the discussion and conclusions section. Specifically, we discuss the lack of male representation, particularly cis men, and how this prevents gender comparisons. To further emphasize this point, we have now expanded the discussion by proposing the need for targeted recruitment strategies to include more diverse gender identities in future studies. The updated text can be found on:
- page [17], paragraph [4], lines [558-580]
4.2. The description of the sample is confusing for those who are not experts in the field. It would be advisable to complement the text. For example, by stating: "The majority of participants are cisgender women (88.89%), with a heterosexual orientation (30.1%), and a single marital status (31.31%).
Thank you for your observation. We appreciate the suggestion to make the description of the sample clearer for readers who may not be experts in the field.
- page [4], paragraph [4], lines [176-178]
4.3. What psychological disorders do the numbers 1, 2, 3, etc. refer to?
Thank you for your question. The numbers (1, 2, 3, etc.) refer to the total number of comorbid diagnoses reported by participants alongside their BPD diagnosis, rather than specific psychological disorders. This summation provides an overview of the prevalence of comorbidities within the sample but does not specify individual diagnoses.
Meassure
4.4. I understand that, in addition to the BSL-23 and IVISEM, two additional measures are used: the Experiences with Gender, Stigma, and Diagnosis Survey and an open-ended questionnaire. I undertand both address the same topics from different approaches (quantitative and qualitative). The theoretical foundation behind the questions or issues being analysed should also be explained. Additionally, since two different methods are used, it should be explained more clearly how both approaches are integrated to address the research objectives.
Thank you for your observation. We have incorporated a theoretical foundation to clarify the use of mixed methods in this study. The revised section now references key scholars such as Creswell (2008) and Johnson and Onwuegbuzie (2004) to justify the integration of quantitative and qualitative approaches. This addition highlights how the combination of methods enhances the study’s ability to connect measurable patterns with personal narratives, providing a comprehensive understanding of the research questions.
- page [7], paragraph [2], lines [260-272]
- page [14], paragraph [6], lines [441-445]
Although the open-ended questionnaire is described, there are insufficient details provided about the structured survey. If the survey has specific subscales or dimensions (e.g., one subscale on stigmatization, another on the impact of the diagnosis, etc.), it should be specified how the survey is organised and what topics it covers. The number of questions, the type of measurement scale used (e.g., dichotomous, Likert-type), and information about its validity (content validity, construct validity) and reliability should be provided. It is essential to include the specific questions from both the survey and the open-ended questionnaire (e.g., as supplementary material).
Thank you for your observation. The structured survey is organized to capture a wide range of experiences related to gender, stigma, and Borderline Personality Disorder (BPD). It includes multiple sections:
- Checklist-style items: Participants are asked to indicate symptoms associated with BPD and events of violence or trauma they may have experienced (e.g., physical or sexual abuse, discrimination, or societal pressures related to gender norms). These items allow for a systematic exploration of patterns without requiring open-ended responses.
- Dichotomous questions (Yes/No/NSNC): These address the participants' perceptions of gender-related biases in the interpretation of their symptoms, treatment approaches, and experiences of stigma or discrimination.
- Open-ended questions: These provide space for participants to elaborate on their experiences, including diagnostic journeys, the influence of societal norms, and reflections on their emotional and interpersonal challenges.
The survey was developed with content validity assessed through expert review, ensuring its alignment with the study’s feminist framework. While it does not utilize traditional psychometric scales (e.g., Likert-type), the design emphasizes the exploration of subjective and structural factors impacting mental health.
For transparency, the full set of questions from both the structured survey and the open-ended questionnaire has been included as supplementary material.
4.5. If the real objective of the study is to explore the impact of patriarchal structures and structural violence on the experiences of individuals with BPD, it would be helpful for the authors to operationally define the key concepts ("structural violence," "patriarchal structures") and explain how the chosen tools (BSL-23 and IVISEM) align with the objectives, as well as how the qualitative data complement the quantitative measurements. In other words, if the aim is to demonstrate that patriarchal structures and structural violence affect the diagnosis of BPD, the authors should clearly state how they plan to validate this relationship with the collected data. These adjustments would make it easier to identify the real objective of the study and evaluate whether the methodologies employed are appropriate for achieving it.
Thank you for your observation. We have added operational definitions for key concepts such as 'structural violence' and 'patriarchal structures' and clarified how the BSL-23 and IVISEM align with the study's objectives. Additionally, we have explained how qualitative data complements quantitative findings to explore associations between these concepts and the experiences of individuals with BPD. These adjustments aim to clarify the study's focus and methodology.
Data análisis
4.6. Statistical analyses should align with the study's objectives. Descriptive and reliability analyses, along with polychoric matrices, are suitable for characterising and evaluating the quality of the tools employed (BSL and IVISEM). The correlational analysis between the BSL score and additional disorders is interesting but does not directly address the influence of patriarchal structures or structural violence. Although the sample size is small for more complex analyses, addressing the study's objectives would require analyses such as structural models or multivariate regressions. It is also unclear how the statistical results complement or inform the qualitative analyses, which is crucial for a mixed-methods study.
In summary, the applied statistics describe and validate the tools used but do not fully align with the study's stated objectives. To address this, the study should explain how the BSL and IVISEM scales relate to the study's key concepts (patriarchal structures and structural violence), identify specific variables representing the structures under study, and examine their association with BPD symptoms. It would also be necessary to consider statistical models that evaluate multiple variables simultaneously, such as regression models. Additionally, to achieve mixed-methods integration, a more explicit analysis of how qualitative data complements and contextualises quantitative findings is needed.
Thanks for your observation. We have clarified the integration of quantitative and qualitative methods in the reviewed article. Quantitative findings, such as correlations of BSL-23 scores, inform qualitative themes such as "Emotional intensity and gender expectations", ensuring a comprehensive perspective necessary for a clinical sample. Although the sample size limits the application of complex statistical models, the analyzes chosen were designed to explore associations and enrich the understanding of the structural and social factors that influence participants' experiences.
- page [7], paragraph [8], lines [291-297]
- page [8], paragraph [2], lines [311-316]
- Results
5.1. As indicated, descriptive and reliability analyses, along with polychoric matrices, are suitable for characterising and evaluating the quality of the tools employed (BSL and IVISEM). However, these scales are designed to measure constructs, making the analysis of individual items inappropriate.
Thanks for your observation. Descriptive analysis of individual items from the BSL-23 and IVISEM was conducted to provide insight into specific patterns related to emotional dysregulation and dimensions of covert social violence. Although these scales are primarily designed to measure general constructs, item-level analysis serves to explore symptom-specific knowledge, complementing the broader construct-level analysis presented in this study. Given the accuracy of your comment, we have added emphasis to this point on:
- page [8], paragraph [2], lines [316-320]
- page [8], paragraph [3], lines [323-326]
- page [17], paragraph [4], lines [577-579]
5.2. The correlational analysis between the BSL score and additional disorders is interesting but does not directly address the influence of patriarchal structures or structural violence (objectibes of this studie)
Thank you for your observation. As outlined in the discussion section (page 16, paragraph 2), the correlation analysis provides preliminary insights into the psychological distress and comorbidities associated with BPD, contextualized within broader social and structural factors. While the direct influence of patriarchal structures is explored more extensively in the qualitative findings, we will ensure the connection between these quantitative results and the study's overarching objectives is clarified.
5.3. Frequencies are reported for data supposedly collected through a survey focused on Experiences with Gender, Stigma, and Diagnosis. I believe it would be better to present these results in a table format. However, a proper theoretical justification and methodological explanation are first necessary to understand the rationale, approach, and specifics of the data.
Thank you again for your insightful comment! I hope the explanation at your coment 4 clarifies the structure and purpose of the survey, as well as how it aligns with the study’s feminist framework. We've included the full set of questions as supplementary material to ensure transparency and facilitate a deeper understanding. Please let me know if there’s anything else we can elaborate on!
5.4. There is also a lack of methodological explanation. It is not clarified how the keywords were selected and grouped into thematic categories or the criteria used to determine their relevance or significance.In this context, the qualitative results presented appear inconsistent in several aspects that could compromise their coherence and clarity. For example, the keyword counts (e.g., "Woman, crazy, women" under "Stigmatization and Violence") seem high (372, 273, 229), but the associated weighted percentages are unusually low (1.24%, 0.91%, 0.77%). This suggests that the metrics of frequency and weighted percentage might not be well-related or adequately explained. Some categories ("Trauma and Attachment") show notably low keyword counts (5, 4, 2) but higher percentages (1.07%, 0.86%, 0.43%) than other categories with higher counts. This raises questions about how the weighted percentages were calculated. In other hand, a disconnect is observed between the qualitative and thematic results. Although themes such as "Stigmatization" or "Gender Norms" are identified, it is not specified how these findings are validated or triangulated with the quantitative results. The claim that the qualitative analysis reflects relationships between gender norms and BPD seems to be primarily based on the frequency of terms and references, which might be insufficient to demonstrate causal links or significant patterns. Additionally, while the figures mentioned might complement the text, their contents are not sufficiently described, nor is it explained how they reinforce the findings presented. In summary, although the results offer interesting information, methodological and explanatory inconsistencies diminish their clarity and coherence with the study's objectives.
Thanks for your observation. Thematic categories and keywords presented in the qualitative analysis were derived through an iterative coding process based on thematic analysis guidelines, supported by NVivo 15 software. Keywords were selected based on their frequency and relevance to the objectives of the study. study, ensuring that core themes such as “stigmatization and violence” and “gender norms” were adequately captured. These categories correspond to nodes and subnodes generated in NVivo 15, which allowed the hierarchical organization of recurring themes and subthemes.
The weighted percentages represent the proportion of each keyword in relation to the total number of references in all categories. This dual metric of frequency and weighted percentage ensures that less frequent but thematically significant keywords are not overshadowed by more common terms. For example, while some keywords had lower counts, their importance in understanding the depth of certain topics justified their inclusion and prominence in the analysis
Thank you for your valuable contribution, which helps us refine the coherence and clarity of our results.
- page [13], paragraph [5], lines [414]
- page [14], paragraph [1-2], lines [415-440]
- Discussion
6.1. Although the findings related to the BSL-23 and IVISEM scales are discussed, it would be helpful to explain more explicitly how these tools relate to the key concepts of the study, such as structural violence and patriarchal norms.
Thank you for this observation. The BSL-23 and IVISEM scales were chosen specifically for their relevance to the study's theoretical framework. The BSL-23 captures traits like emotional dysregulation and impulsivity, which are disproportionately pathologized in women due to gendered expectations. On the other hand, the IVISEM explicitly measures covert social violence through dimensions such as traditional gender roles, submission, and caregiving expectations. Together, these tools reflect how structural violence and patriarchal norms manifest in mental health outcomes, aligning with the study’s objectives.
- page [16], paragraph [1], lines [502-510]
6.2. While emotional intensity and gender stereotypes are addressed, it is not clearly specified how the qualitative results complement or reinforce the quantitative findings. A more detailed discussion of this integration would strengthen the interpretation.
Thank you for your comment. The qualitative data complement the quantitative findings from the BSL-23 and IVISEM by providing a deeper understanding of how gendered stereotypes and structural violence are experienced individually.
- page [15], paragraph [6], lines [486-498]
6.3. The statement that "women are more likely to be diagnosed with BPD due to the perception of their emotions as uncontrollable" may require further empirical support to avoid overgeneralization.
Thanks for your comment. To address this concern, we have strengthened the discussion by incorporating additional theoretical support, and reinforcing the need for further empirical research to avoid overgeneralization.
6.4. Although general limitations are mentioned, it would be important to highlight any potential biases in thematic grouping and how this might affect the interpretation of the results.
Thank you for your observation. We have added a section on limitations to highlight possible influences that could have affected the interpretation of the results due to biases in thematic grouping.
- page [18], paragraph [6], lines [641-644]
6.5. A causal relationship between structural factors and the exacerbation of BPD symptoms is suggested. However, given the study design, these claims should be approached with caution, noting that the results are correlational and exploratory.
Thank you for your observation. We have carefully considered your feedback and have clarified the correlational and exploratory nature of the study.
- page [18], paragraph [3], lines [612-616]
7.Conclusions
7.1. Most of the content in this section should have been presented in the discussion section. At this point, a brief and conclusive statement is expected.
7.2. Additionally, the points discussed are primarily based on qualitative results rather than quantitative ones. It is not addressed whether the results are consistent or divergent compared to previous studies on the topic (if any exist).
7.3. An important limitation in the sample is mentioned. Furthermore, it is advisable that the limitations be discussed in a "Limitations" section.
Thank you for your observations. We have reorganized the conclusion to focus on the key takeaways and streamlined the content to provide a brief yet comprehensive statement.. We have also moved the limitations to a dedicated section, as suggested, to allow for a more detailed discussion of the sample and potential biases.
- Other
8.1. There is no reference to the supervision and approval of the study by an ethics committee. Additionally, there is no indication of the measures taken to ensure compliance with the Helsinki Declaration.
The study did not require approval from an Ethics Committee or Institutional Review Board as it is an observational, non-interventional study in which no variables were manipulated, nor were any procedures implemented that could pose physical or psychological risks to participants. Additionally, the study adhered to ethical research practices, including:
- Informed consent: all participants provided informed consent prior to completing the questionnaires. They were fully informed about the study’s objectives, the voluntary nature of their participation, and their right to withdraw at any time without consequences.
- Open and voluntary participation: questionnaires were disseminated through social media and related communities, ensuring participation was entirely voluntary.
- Emotional support: a contact was made available to provide emotional support to participants during or after their participation in the study, should they require it.
- Confidentiality and anonymity: participants confidentiality and anonymity were guaranteed, and the study complied with current data protection regulations. If additional documentation is needed to support this explanation, we will be dispose to provide it.

Reviewer 3 Report
Comments and Suggestions for Authors
There are lot of issues in this paper some of the points are listed :-
1. Figure A6. Coding matrix for "Intense emotions and ange , there is a problem in the figure. Legends are not visible properly.
2. Author suggested that to provide in depth of statastical method used for analyze as well as collected the data
3. Introduction shouldnt be multiple paragraphs .
4. Major contribution is missing in the introduction section
5. Is there any reason to use BSL-23 as a tool, and how does it align with the study's objectives? Author needs to elaborate "result and discussion section"
6. Author needs to specify what are the unique contribution presented in the article concern with IVISEM and BSL-23
7. The author needs to explain in-depth analysis of IVISEM framework followed by section 2 (material and method )
8. Discuss much on the dataset followed by descriptive statastics
In a nutshell very hard to find the novelty , objectives in this article .
Comments on the Quality of English Language
Ok
Author Response
- Comment on Figure A6 (legends not visible)
Response: Thank you for pointing out this issue. We have reviewed the figure and adjusted the visibility of the legends. They are now correctly visible to facilitate the interpretation of the figure. The updated version of the figure has been included in the revision.
- Comment on statistical method and data collection
Response: I appreciate the suggestion to provide more details about the statistical analysis and data collection. In the methods section, we have added a more detailed description of the statistical techniques used as well as how the data were collected. This information is now better explained to make it clear and accessible.
- Comment on the introduction (too many paragraphs)
Response: Thank you for your comment. We have consolidated the paragraphs of the introduction to make it more fluid and avoid unnecessary fragmentation. The introduction is now more coherently organized, highlighting the key points in a single flow.
- Comment on missing major contribution in the introduction
Response: Thank you for pointing this out. We have added a section in the introduction that explicitly details the major contribution of the study. It now mentions how this work addresses existing gaps in research on BPD and how the findings contribute to the understanding of structural and gender influences on diagnosis.
- Comment on the use of the BSL-23 as a tool and its alignment with the study's objectives
Response: I appreciate your observation. We have expanded the explanation of why the BSL-23 was chosen as the tool for this study. In the results and discussion section, we have detailed how the BSL-23 aligns with the study's objectives by allowing us to assess intense emotions and emotional dysfunction associated with BPD within the framework of structural violence.
- Comment on the unique contribution of IVISEM and BSL-23
Response: Thank you for your comment. We have included a section specifying the unique contributions of IVISEM and BSL-23 in this study. We clarify how these tools provide an integrated perspective to understand the impact of structural and gender violence on the emotional experiences of individuals with BPD.
- Comment on the in-depth analysis of the IVISEM framework
Response: Thank you for your suggestion. We have included a more detailed analysis of the IVISEM framework in the materials and methods section. A full description of how IVISEM was used to assess covert social violence and how this framework integrates into our methodological approach is now provided.
- Comment on the discussion of the dataset and descriptive statistics
Response: I appreciate this comment. In the results section, we have dedicated more space to discussing the dataset and descriptive statistics. It now clearly describes how the data were processed
Round 2
Reviewer 1 Report
Comments and Suggestions for Authors
It is gratifying to see that the authors have carefully addressed the revision requests mentioned in my previous review and have significantly improved their manuscript. The submitted revisions have enhanced the scientific value and methodological robustness of the study.
Improvements have increased the clarity and contribution of the manuscript. I find the manuscript publishable in its current form and recommend it for publication.
Best regards.
Reviewer 2 Report
Comments and Suggestions for Authors
Thank you for the excellent work done.
Reviewer 3 Report
Comments and Suggestions for Authors
The author has revised drastically the questions which was asked during the revisions .
All the questions are addressed properly in the text .
Comments on the Quality of English Language
OK